

# Possible provenance of IRD by tracing late Eocene Antarctic iceberg melting using a high-resolution ocean model

Mark V. Elbertsen[1], Erik van Sebille[2, 3], and Peter K. Bijl[1]

[1]Department of Earth Sciences, Utrecht University, Princetonlaan 8a, 3584 CB Utrecht, The Netherlands
[2]Institute for Marine and Atmospheric Research Utrecht, Department of Physics, Utrecht University, Princetonplein 5, 3584 CC Utrecht, The Netherlands
[3]Centre for Complex System Studies, Utrecht University, Leuvenlaan 4, 3584 CE Utrecht, The Netherlands

**Correspondence:** Mark V. Elbertsen (m.v.elbertsen@uu.nl)

**Abstract.** The Eocene-Oligocene Transition is characterised by the inception of the large-scale Antarctic ice sheet. However, evidence of earlier glaciation during the Eocene has been found, including the presence of ice-rafted debris at Ocean Drilling Program Leg 113 Site 696 on the South Orkney Microcontinent (Carter et al., 2017). This suggests marine-terminating glaciers should have been present in the southern Weddell Sea region during the late Eocene, generating sufficiently large icebergs to
South Orkney to survive the high Eocene ocean temperatures. Here, we use Lagrangian iceberg tracing in a high-resolution eddy-resolving ocean model of the late Eocene (Nooteboom et al., 2022) to show that icebergs released from offshore the present-day Filchner Ice Shelf region and Dronning Maud Land could reach the South Orkney Microcontinent during the late Eocene. The high melt rates under the Eocene warm climate require a minimum initial iceberg mass on the order of $100\,\mathrm{Mt}$ and an iceberg thickness of several tens of metres to be able to reach the South Orkney Microcontinent. Although these sizes are at
the larger end of the present-day range of common iceberg sizes around Antarctica, the minimum estimates are not unfeasible and, hence, the present study confirms previous findings suggesting glaciation and iceberg calving were possible in the late Eocene.

## 1 Introduction

A period of long-term global cooling through the middle and late Eocene, interrupted by the relatively short but warm Middle
Eocene Climatic Optimum (MECO; $40\,\mathrm{Ma}$), eventually set the stage for the formation of a continental-scale Antarctic ice sheet around the Eocene-Oligocene Transition (EOT) at roughly $34\,\mathrm{Ma}$ (Hutchinson et al., 2021) as $CO_2$ concentrations declined below threshold levels required for ice formation (DeConto and Pollard, 2016; Pagani et al., 2011). The Antarctic continent is generally considered to be largely ice-free until the end of the Eocene (Singh et al., 2022; Zachos et al., 2001) when an abrupt increase of $1.2$ to $1.5\,‰$ is observed in the global marine benthic $\delta^{18}O$ record (Miller et al., 1987; Tigchelaar et al., 2011),
which, after removing the temperature component, corresponds to an increase in global ice volume equivalent with 60 to 130 % of the modern East Antarctic Ice Sheet volume (Bohaty et al., 2012).

However, multiple proxy-based studies have shown evidence for the presence of ice on Antarctica before the EOT, such as around $37.5\,\mathrm{Ma}$ (Scher et al., 2014), $36\,\mathrm{Ma}$ (Peters et al., 2010) and $36.5\,\mathrm{Ma}$ (Carter et al., 2017). This latter study is based on





the finding of significant amounts of fine-grained ice-rafted debris (IRD) at Ocean Drilling Program (ODP) Leg 113 Site 696, taken on the southeastern margin of the South Orkney Microcontinent (SOM; Fig. 1). As iceberg calving only occurs when glaciers are marine-terminating (Diemand and Dryak, 2019), this suggests that vast amounts of ice were already present at this time (Carter et al., 2017). In turn, this would require sufficient snow supply to Antarctica to maintain the mass balance under the still relatively high Eocene temperatures (Baatsen et al., 2024; Douglas et al., 2014; Thompson et al., 2022; Nooteboom et al., 2022), although these high temperatures themselves could have led to increased precipitation on Antarctica (following

the Clausius-Clapeyron relation; Le Treut and Ghil, 1983). In combination with a suitable topography, creating sufficiently low temperatures at higher altitudes for snowfall to survive the summers, this could have allowed glacier formation. Still, such high temperatures would imply that icebergs, if present, would have been relatively short-lived (Carter et al., 2017) or relatively large in size, thereby lengthening their lifetime and allowing them to travel larger distances (Wesche and Dierking, 2014). In addition, if glaciers were present in the late Eocene, the local temperature might be somewhat overestimated as the

fully-coupled climate models do not include ice-climate interactions.

Nevertheless, the formation of a substantial mass of ice before the EOT would complicate the mass balance of $\delta^{18}$O as this should have led to an earlier positive shift in oxygen isotopes, which is not obvious in the record (Bohaty et al., 2012) and does not fit sea level estimates (Wilson et al., 2013). Most modelling studies also do not support such large-scale glaciations (Baatsen et al., 2024) although a recent study using a novel ice sheet-climate modelling approach showed that large-scale,

though short-lived, glaciations might have been possible when $CO_2$ levels were sufficiently low and coincided with summer insolation minima (Van Breedam et al., 2022), such as during the Priabonian oxygen isotope maximum (PrOM) at 37 Ma (Scher et al., 2014). In addition, ice-climate interactions are not included in these models, which could thus lead to a much warmer local environment than with ice present.

From the geochronology analysis of the IRD from ODP Site 696 and comparison to the Antarctic continental geology,

Carter et al. (2017) indicated the southern Weddell Sea region as the likely source region, whilst a contribution from the nearby Antarctic Peninsula was excluded. Hence, this implies that local ocean currents in the late Eocene should have been relatively similar to today in order to bring the icebergs from the southern Weddell Sea region to ODP Site 696 at the SOM (Fig. 1), next to the icebergs being large enough in size to survive the flow path from Antarctica to ODP Site 696. Currently, the SOM is located in the so-called Iceberg Alley, where most icebergs formed at the Antarctic continent float equatorward (Carter et al.,

2017). Indeed, such a local circulation in the Eocene Weddell Sea would be consistent with information from microfossil assemblages (Bijl et al., 2011; Sauermilch et al., 2021) and model simulations (Huber et al., 2004; Sijp et al., 2011).

Therefore, this study aims to reconstruct possible Antarctic iceberg trajectories in the late Eocene and compare these to proxy evidence found at ODP Site 696 by adding iceberg melt parametrisations to Lagrangian tracks of floating particles in an Eocene ocean model. Although reconstructions of late Eocene temperature gradients cannot generally be reconciled with

plankton-based circulation patterns in current numerical models (Baatsen et al., 2020), a high-resolution ($0.1°$) eddy-resolving ocean model of the late Eocene was found to agree better with the circulation patterns and temperature estimates from proxies (Nooteboom et al., 2022), making this model potentially suitable for analysing whether icebergs could have brought IRD to ODP Site 696 at the SOM. If the IRD is indeed derived from Antarctica, the simulations must demonstrate that 1) icebergs



derived from the southern Weddell Sea region can indeed reach ODP Site 696 before having melted away, 2) which regions are most likely to have released icebergs, and 3) what the minimum size of the icebergs leaving Antarctica must be in order to survive the flow path from Antarctica to ODP Site 696. Otherwise, the question arises of what causes the disagreement between the IRD and model data, for example, a difference in circulation or too high temperatures for icebergs to survive.

## 2 Methods

### 2.1 Iceberg model

In this study, icebergs are traced in a Lagrangian framework using Parcels (Delandmeter and van Sebille, 2019). By adding iceberg melt parametrisations, the size-dependent trajectories of icebergs can be modelled as the size of an iceberg - and consequently, the depth it reaches - changes with time and, hence, determines the influence of the horizontal circulation on the iceberg drift. Note that in this model, the depth-integrated iceberg drift is forced solely by the ocean velocity field so that the influences of wind drag, the Coriolis force, wave radiation force and pressure force are ignored (Supporting Information S2.1.4).

#### 2.1.1 Iceberg shape and size

Icebergs exist in a wide range of shapes and sizes that are always irregular to at least some extent. To simplify the simulations, we follow previous literature (e.g., Rackow et al., 2017; Martin and Adcroft, 2010) and approximate the icebergs as cuboids. We assume a length-to-width ratio of $L : W = 1.5 : 1$, consistent with modern observations (Bigg et al., 1997). The thickness or height $H$ of an iceberg consists of a part above water, the freeboard or sail $F$, and a submerged part, the draft or keel $D$, following $H = F + D$. Moreover, the draft and thickness are related through density as $D = \alpha H$ with $\alpha$ the ratio of the typical iceberg density in the Southern Ocean $\rho_i = 850.0 \ \mathrm{kg \ m^{-3}}$ (Silva et al., 2006) over the average seawater density $\rho_o = 1027.5$ $\mathrm{kg \ m^{-3}}$. Note that the Eocene iceberg density might have been different from the present-day value as it depends on the density of the glacier it calved from, which in turn is influenced by temperature, precipitation rate, wind speed and topography (Ligtenberg et al., 2011). At present, relatively high ice densities are found for glaciers under the influence of high wind speeds, high precipitation rates and high temperatures, suggesting that the high temperatures and precipitation rates of the Eocene (Baatsen et al., 2024) might have caused higher ice densities as well.

For the present-day, iceberg size distributions have been constructed based on (satellite) observations (Wesche and Dierking, 2014; Gladstone et al., 2001). However, these cannot be applied directly to the Eocene as climatic and glacial conditions, such as the temperature and precipitation patterns and glacial extent, differed and, hence, the number and size of released icebergs might have been different. Nevertheless, as we are interested in the possibility of icebergs reaching ODP Site 696, we can define iceberg lengths of several order-of-magnitudes, varying from 10 to 100000 m (Table 1), to obtain a general idea of possible iceberg trajectories.



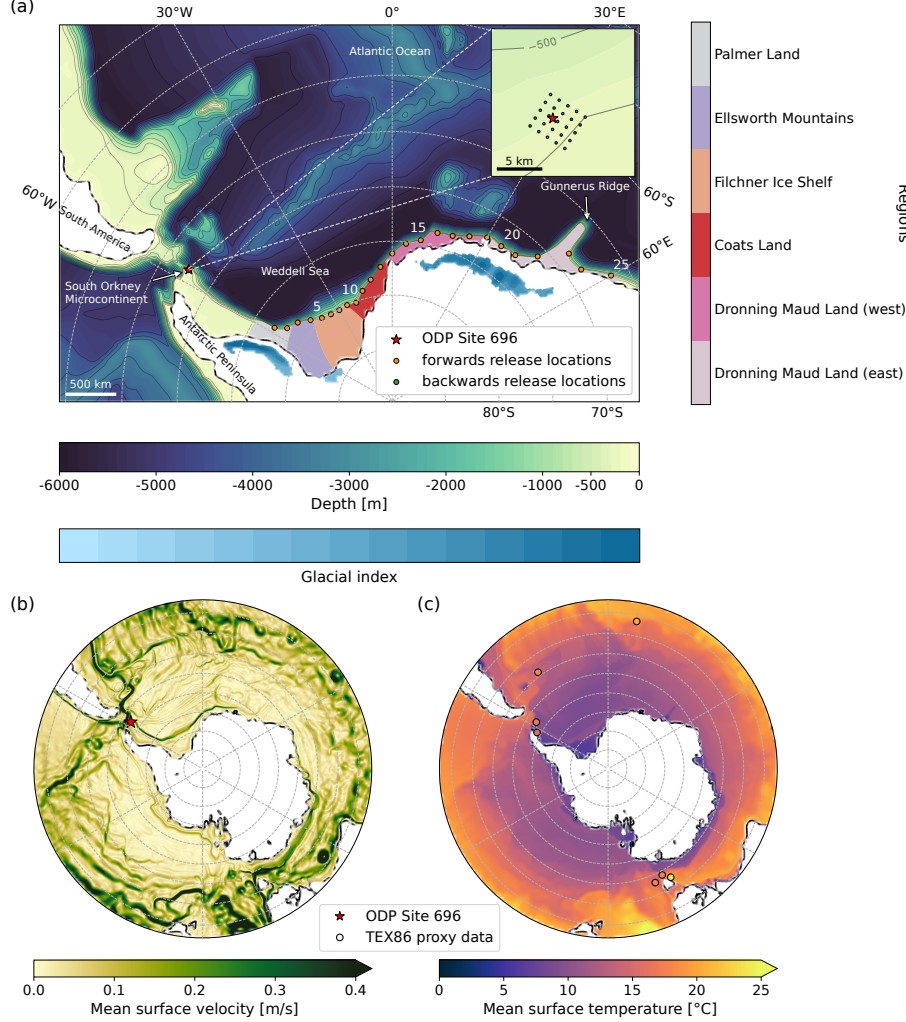

**Figure 1.** Eocene geography of the Weddell Sea region (Nooteboom et al., 2022) with (a) the approximate location of ODP Site 696 and the 25 forwards and 25 backwards iceberg release locations defined in this study. Also shown are the potential provenance regions from Palmer Land to Dronning Maud Land based on Carter et al. (2017) and potential regions of ice formation during the late Eocene as given by a glacial index calculated from a simple surface mass balance by Baatsen et al. (2024). Regional time-mean (model years 38-42) ocean properties from the high-resolution late Eocene model fields by Nooteboom et al. (2022) are given for the (b) velocity in the surface layer and (c) temperature at the surface ($\sim$ 0-200 m). The latter figure includes TEX$_{86}$ surface ($\sim$ 0-200 m) temperature data from the late Eocene for ODP Sites 1172, 1170, 1090, 1168, and 696, DSDP Site 511 and Seymour Island. Note that the marker of ODP Site 696 is roughly a quarter of the size of the SOM in panel (a) or roughly equal to the size of the SOM in panel (b).

The initial thickness of the icebergs is defined such that $L : H = 5 : 1$, which is a common ratio for several iceberg types, including tabular icebergs (Turnbull et al., 2015). However, for the smallest size class ($L = 10$ m), this would give a thickness



**Table 1.** Definition of the initial iceberg size classes used to simulate late Eocene Antarctic icebergs in this study. The iceberg length is defined for several orders-of-magnitude and the iceberg width follows from assuming icebergs are cuboids with a length-to-width ratio of 1.5. A length-to-thickness ratio of 5 is used with a cut-off thickness of 500 m. Finally, the iceberg mass is calculated using an ice density of 850.0 kg m$^{-3}$. Note that the size of class C1 is adapted slightly to fit the ocean model layers.

| Class | L [m] | W [m] | H [m] | M [Mt] |
|-------|-------|-------|-------|--------|
| C1 | 17 | 11.3 | 12.2 | $2.00 \times 10^{-3}$ |
| C2 | 100 | 66.7 | 20 | $1.13 \times 10^{-1}$ |
| C3 | 1 000 | 666.7 | 200 | $1.13 \times 10^{2}$ |
| C4 | 10 000 | 6 666.7 | 500 | $2.83 \times 10^{4}$ |
| C5 | 100 000 | 66 666.7 | 500 | $2.83 \times 10^{6}$ |

of 2 m and a draft of only 1.65 m. As the base of the uppermost ocean layer of the model lies at 10.01 m and iceberg motion is only determined by ocean flow, this would remove the basal melt term and hence keep the iceberg thickness static. Therefore, instead, we set the iceberg thickness to 12.2 m ($D = 10.09$ m). Note that this also requires a change in iceberg length to suffice the tipping criterion explained below.

A second adaptation is required for the maximum iceberg thickness, which is constrained by the thickness of the ice shelf front the berg calved from (England et al., 2020; Gladstone et al., 2001). Whilst most authors use a maximum thickness between 250 and 350 m, related to the average modern ice shelf front thickness (England et al., 2020; Gladstone et al., 2001), here we use a maximum iceberg thickness of 500 m as the warm Eocene monsoonal climate at the Antarctic coast (Baatsen et al., 2024) could potentially lead to high snow accumulation rates and ample meltwater, creating fast-flowing ice-sheets and glaciers with

warm bases. In the present day, such ice sheets and glaciers have been found to generate thick icebergs in both Antarctica (Dowdeswell and Bamber, 2007) and Greenland (Dowdeswell et al., 1992). In addition, the present-day Filchner Ice Shelf produces icebergs over 500 m in thickness due to its constrained topographical setting (Dowdeswell and Bamber, 2007) and thus might have done so in the past.

Throughout the duration of the run, the length-to-width ratio of the icebergs is kept constant. The ratio between the iceberg

area and thickness, however, changes through the difference in melt rates at the iceberg sides and base. If the width-to-height ratio $W/H$ becomes smaller than a critical value, $\varepsilon_c$, the iceberg capsizes, and the iceberg width and thickness are interchanged. Here, $\varepsilon_c = \sqrt{6\alpha(1-\alpha)}$ with $\alpha = \rho_i/\rho_o$ as adapted by Wagner et al. (2017a) from Weeks and Mellor (1978). Inclusion of this effect can significantly lengthen the lifetime of especially small ($L < 500$ m) icebergs (Wagner et al., 2017a).

### 2.1.2 Iceberg deterioration

Along its trajectory, the size of an iceberg changes through several processes at both the ice-water and ice-air interface. Following previous literature (e.g., Martin and Adcroft, 2010; Gladstone et al., 2001; Rackow et al., 2017), we limit the included





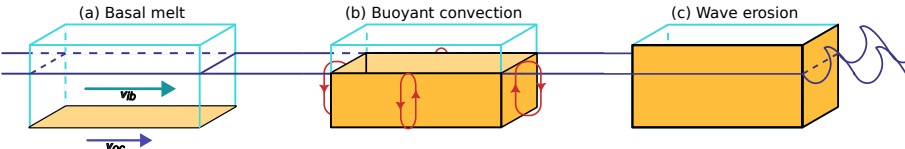

**Figure 2.** The three processes influencing iceberg deterioration included in the iceberg model. The coloured areas indicate the iceberg areas on which the melt process works. (a) Basal melt is caused by the turbulence generated by the relative motion of the iceberg ($v_{ib}$) in the ambient seawater ($v_{oc}$). Note that this is the only process affecting the iceberg base. (b) Buoyant convection arises due to the movement of relatively warm ocean water along the submerged iceberg sides. (c) Wave erosion occurs through the impact of relatively warm ocean waves on two iceberg sides.

processes in this study to basal melt, buoyant convection and wave erosion, which are the main contributors to iceberg melting (Bigg et al., 1997) and were found to cause $98\%$ of the total melt in the Arctic (El-Tahan et al., 1987).

Firstly, turbulence generated by the relative motion of the iceberg in the ambient seawater leads to forced convection or basal melt $M_b$ at the iceberg base (Fig. 2a). This term is especially important for giant icebergs (Rackow et al., 2017) and can reach up to $1\,\mathrm{m\,d^{-1}}$ (Bigg, 2015), influencing the iceberg thickness. Secondly, the movement of (warm) ocean water along the submerged sides of the iceberg leads to a heat exchange through buoyant convection $M_v$ (Fig. 2b). This term is usually relatively small, especially in colder areas, with a maximum of roughly $0.2\,\mathrm{m\,d^{-1}}$ (Cenedese and Straneo, 2023), influencing the horizontal iceberg extent. However, the warmer Eocene ocean might increase the contribution of this term. Thirdly, the impact of waves of relatively warm water causes the iceberg to lose mass around the waterline through wave erosion $M_e$ (Fig. 2c). This term usually contributes most to iceberg melt and can easily reach $1\,\mathrm{m\,d^{-1}}$, even for small waves (Cenedese and Straneo, 2023) and also influences the horizontal iceberg extent. A more detailed description of the melt terms, including their formulation, can be found in previous literature (e.g., Martin and Adcroft, 2010; Merino et al., 2016) and the Supporting Information S2.1.3. However, we will shortly focus on adaptations made here.

Although many studies use ocean surface properties to calculate the above melt terms (Martin and Adcroft, 2010; Marsh et al., 2015), we follow Merino et al. (2016) and use depth-integrated values along the iceberg keel. A second change with respect to much previous literature is the adaptation of the internal ice temperature, which is generally set to $-4\,°\mathrm{C}$. However, the use of the freezing temperature instead of the internal ice temperature was shown to give better results (FitzMaurice and Stern, 2018). Therefore, we use a value of $T_f = -1.92\,°\mathrm{C}$ (as derived in the Supporting Information S2.1.3). Finally, iceberg grounding occurs when the iceberg draft exceeds the water depth. In this case, the depth-integrated iceberg velocity is set to zero for the advection (Supporting Information S2.1.4), inhibiting the iceberg from moving further until it has melted sufficiently during the next timestep(s), i.e. when the iceberg draft is smaller than the water depth. Note that this can only be realistically implemented in forward runs (Sect. 2.3).





### 2.1.3 Notes on the iceberg model

Several choices and assumptions have been made while building the iceberg model. Firstly, the approximation of icebergs as cuboids can lead to an underestimation of melt rates and, hence, an overestimation of the iceberg lifetimes due to the reduced areas (Hester et al., 2021). In addition, the length-to-width ratios of icebergs are more variable in reality, both through time and at release (Weeks and Mellor, 1978). Still, for most icebergs, the ratio varies between $1:1$ and $2:1$ (Weeks and Mellor, 1978). In addition, a process similar to basal melting occurs along the submerged iceberg sides (Bigg et al., 1997) and is sometimes

also included in models (Rackow et al., 2017). However, as this melt term depends on the friction velocity between the ocean flow along the iceberg keel and the iceberg motion itself, this requires iceberg dynamics not solely dependent on ocean velocity as used in our approach, otherwise the friction velocity will always be zero. Therefore, this melt term was not included in our model.

Most uncertain, however, is the inclusion of wave erosion. The general form of the equation includes a damping term related

to sea-ice cover (Supporting Information S.2.1.3). Although no direct information on sea-ice cover exists in the Eocene model used, we can use sea surface temperatures as an indication of the possibility of sea ice formation. Assuming sea ice only forms below $-1.8\,^{\circ}\mathrm{C}$ (Baatsen et al., 2020) shows no potential for sea ice formation as temperatures are constantly higher. We can, therefore, assume that the dampening effect of sea ice on wave erosion was practically absent. This is also supported by evidence from dinoflagellate cysts suggesting sea ice formation started during the Oligocene (Houben et al., 2013).

Even more important for wave erosion is the representation of the wind. Whilst all other input fields are given at a daily resolution (Sect. 2.2), we have to restore to monthly wind stress fields as no wind velocity fields are available, and neither exists on a daily resolution. Comparison using high- and low-resolution temporal data of the present-day shows this leads to an average difference in wave erosion of 31 % (Sect. 4.4 and Supporting Information S3.1.4). Hence, we expect wave erosion melt rates in this study to be underestimated.

At last, more general exclusions from the model are the computation of weighted means when icebergs extend through multiple grid cells (Rackow et al., 2017) and the parametrisation of iceberg breakup. Exclusion of the latter has been found to overestimate iceberg lifetimes compared to observations (England et al., 2020) and iceberg breakup in open waters can cause up to 80 % of the total iceberg mass loss of large icebergs at present (Tournadre et al., 2015). To avoid this problem, Wagner et al. (2017b) removed such large icebergs from the model run after one year, although this leads to an underestimation of the

icebergs' lifetime as they would not necessarily have disappeared after one year in reality. As such, we do not remove large icebergs from the model but need to keep in mind that their lifetime is likely to be overestimated.

### 2.2 Eddy-resolving ocean model

The simulations in this study are performed using the high-resolution eddy-resolving model of the late Eocene by Nooteboom et al. (2022). This ocean-only Parallel Ocean Program model (POP) (Smith et al., 2010) has a horizontal resolution of $0.1\,^{\circ}$,

which allows it to include smaller-scale structures such as ocean eddies. It is forced at the surface by a fixed atmosphere of the fully coupled simulations of the Community Earth System Model, or CESM, version 1.0.5 (Baatsen et al., 2020) under a $2\times$





preindustrial $CO_2$ forcing to simulate the late Eocene. The bathymetry of CESM was interpolated linearly to the high-resolution POP-grid.

As required for the iceberg melt calculations, multiple fields up to 730 m depth were used, covering a five-year time span (model years 38 to 42) at a daily resolution. These include the 3D ocean horizontal ocean velocity component (Fig. 1b) and 3D temperature (Fig. 1b) fields. In addition, for the same model years - albeit on a monthly resolution - the horizontal surface wind stress components were used to approximate wind velocity as the latter was not available from the model.

Although the high resolution of this model allowed a more accurate simulation of paleotemperatures and circulation (Nooteboom et al., 2022), we need to keep in mind that the model has no dynamic ice component and thus does not account for a regional ice-induced cooling (Manabe and Broccoli, 1985). Therefore, if ice were present in Antarctica during the late Eocene, ocean temperatures in the vicinity of these ice masses might have been slightly lower than projected. Although this effect is likely minor, as the currently projected temperatures are vastly above zero, it might have made iceberg survival somewhat more realistic.

## 2.3 Locations and regions

Several locations and regions were defined in constructing this model. Firstly, we need to determine the paleolocation of ODP Site 696 during the late Eocene. Although presently located at $61°50.959'$ S, $42°55.996'$ W on the southeastern margin of the SOM, paleoceanographic reconstructions (López-Quirós et al., 2021) and visual comparison of the present-day and Eocene bathymetry places the paleolocation of ODP Site 696 around $67°5'$ S, $57°$ W (Fig. 1). During the simulations, we assume icebergs coming within the distance of one grid cell ($\sim 11$ km) to be able to deposit IRD at the site unless otherwise indicated.

Secondly, two sets of iceberg release locations were defined. Using the analysis of the hinterland geology by Carter et al. (2017) as a guideline, we spread 25 locations at the 500 m bathymetry line with a longitudinal spacing of $2°$ between the coast offshore the Ellsworth Mountains and Dronning Maud Land (24 points) and a $5°$ longitudinal spacing along the Antarctic Peninsula (1 point; Fig. 1) to test where icebergs released along the Antarctic coast end up. The coastline was further divided into six regions following the division by Carter et al. (2017), but note that we exclude North Graham Land as the bedrock here was shown to be too young compared to the IRD found at ODP Site 696(Carter et al., 2017). In addition, we include Palmer Land and the eastern sector of Dronning Maud Land, which were not included in the analysis of Carter et al. (2017). During backward simulations, for which 25 locations were defined within the grid cell around ODP Site 696 (Fig. 1), these regions were used to locate possible source regions.

## 2.4 Experimental design

Taken together, this leads to eleven different scenarios tested in this study. Firstly, forward runs (i.e., runs with iceberg releases at the Antarctic Margin forwards in time to see where they end up) with grounding are performed for all five size classes (Table 1). In addition, the sensitivity of these runs to various conditions is tested by studying the effect on icebergs of class C4. This is performed by 1) including icebergs within two grid cells distance to ODP Site 696 ($\sim 22$ km), and 2) executing the





model using only surface fields instead of depth-integrated values as is usually done in iceberg models (e.g., Bigg et al., 1997;
Gladstone et al., 2001; Martin and Adcroft, 2010).

For the backward simulations (i.e., those calculating the trajectory backwards in time away from ODP Site 696 towards
Antarctica), only the three lowest size classes are used since we expect icebergs to be relatively small once reaching ODP Site
696 under the influence of the high late Eocene temperatures. In addition, as for the forward simulations, a simulation using
only surface fields is performed for size class C1. Note that none of these scenarios includes iceberg grounding, as information
on the duration an iceberg was grounded at a certain position is lost in backward time.

For each scenario, icebergs are released daily at all respective release locations for the five-year model period, leading to 1823
releases per site. Sampling is performed at an hourly interval in order to satisfy the Courant-Friedrichs-Lewy (CFL) condition
and avoid overshooting cells during advection. Note that the IRD found at ODP Site 696 does not provide information on the
number of icebergs that have reached the site and, as such, we are interested in the possibility that any iceberg reaches ODP
Site 696 from the defined release locations during the simulations. Still, of course, a larger number of icebergs able to reach
ODP Site 696 would increase the likelihood of any icebergs reaching the site.

## 3 Results

We first study the forward trajectories to analyse if icebergs can reach ODP Site 696 for any of the iceberg size classes. From
this, we can infer possible source locations of IRD. Next, by studying the backward trajectories, we can determine the minimum
iceberg size required at each calving site. Finally, the sensitivity of the model to various settings is analysed.

### 3.1 Flow patterns and iceberg trajectories

Irrespective of the simulation, most icebergs seem unable to reach ODP Site 696 (Fig. 3). From the two smallest size classes
(not shown), none of the icebergs arrived at ODP Site 696. The same holds for size class C3, although in this case, some
icebergs can at least reach the northern part of the Antarctic Peninsula. For size class C4, release locations west of Gunnerus
Ridge are - at least in some cases - able to generate icebergs that can reach ODP Site 696. These trajectories will be referred to
as 'successful' in the rest of the text. For C5, interestingly, none of the icebergs seem able to reach ODP Site 696.

Some general information can also be gleaned from the complete set of trajectories. In all three scenarios shown in the top
row of Figure 3, icebergs initially seem to travel relatively close to the coast in a counterclockwise direction. However, for the
smaller icebergs of class C3, icebergs are able to flow far onto the Ronne-Filchner shelf and over the interior of Gunnerus Ridge.
In class C4, these regions are only sparsely crossed by icebergs, and for size class C5, they are nearly devoid of trajectories.
Moreover, the presence of Gunnerus Ridge seems to induce relatively chaotic patterns in the iceberg trajectories in the vicinity
of this region.

After reaching the tip of the Antarctic Peninsula, icebergs head in several general directions. Firstly, icebergs can travel
northeastwards. For C3, however, most icebergs seem to have melted completely before crossing 65° S. For size class C4 and
especially C5, icebergs can survive longer and are able to leave the domain shown here northwards or in some cases float





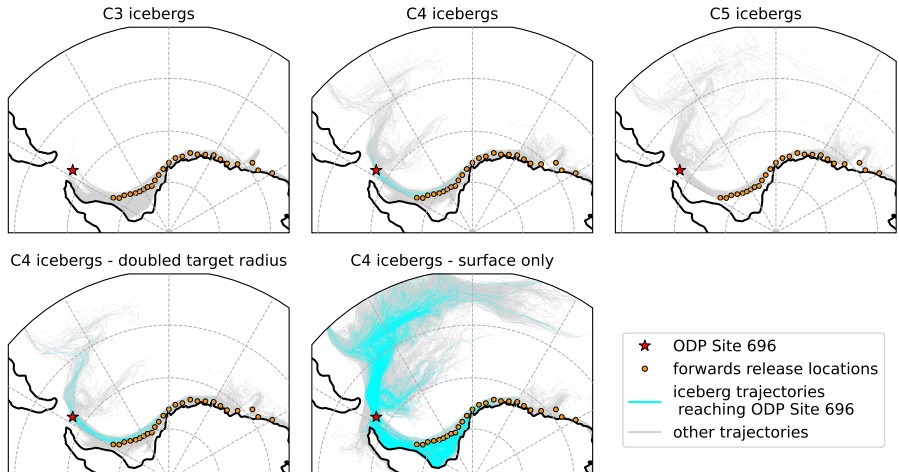

**Figure 3.** Iceberg trajectories from forward simulations that can (blue) or cannot (grey) reach ODP Site 696 within the distance of one grid cell ($\sim 11$ km) at some point along their trajectory. *Top row:* results of the main simulations ran for iceberg size classes C3, C4 and C5. *Bottom row:* results of the sensitivity simulations ran for iceberg size class C4 with inclusion of icebergs passing ODP Site 696 at twice the distance of a grid cell (left) or when using only surface fields (right). The trajectories of size classes C1 and C2 are not shown as none of these trajectories reach near to ODP Site 696. Note that the marker of ODP Site 696 is roughly half the size of the SOM.

eastwards around 67° S. A second route possible after leaving the Antarctic Peninsula starts in the southeastern direction and continues with icebergs heading either northeastward or circling back towards the Antarctic coast.

## 3.2 IRD provenance

To study in more detail where the icebergs reaching ODP Site 696 might have been derived from, we now focus on the percentage of releases from each forward release location that crosses over ODP Site 696 at some point during its trajectory (Fig. 4). As indicated before, none of the release locations lead to successful releases for the smallest three iceberg size classes. For size class C4, however, icebergs can reach ODP Site 696 from several locations along the Antarctic coast. This includes Palmer Land (locations 1 and 2), offshore the Filchner Ice Shelf (locations 5, 7 and 8), and even west and east Dronning Maud Land (locations 13, 15 and 21). For even larger icebergs, no successful trajectories exist. Note that, as such, irrespective of iceberg size, none of the icebergs released offshore the Ellsworth Mountains, Coats Land, nor in the eastern part of Dronning Maud Land east of Gunnerus Ridge are able to reach ODP Site 696 for the main simulations.

## 3.3 Minimum iceberg size

Next, by backtracking icebergs from ODP Site 696 to the Antarctic coast, we obtain a first indication of the minimum iceberg size required at different locations along the Antarctic coast. First, the results are described spatially to visualise the extent of



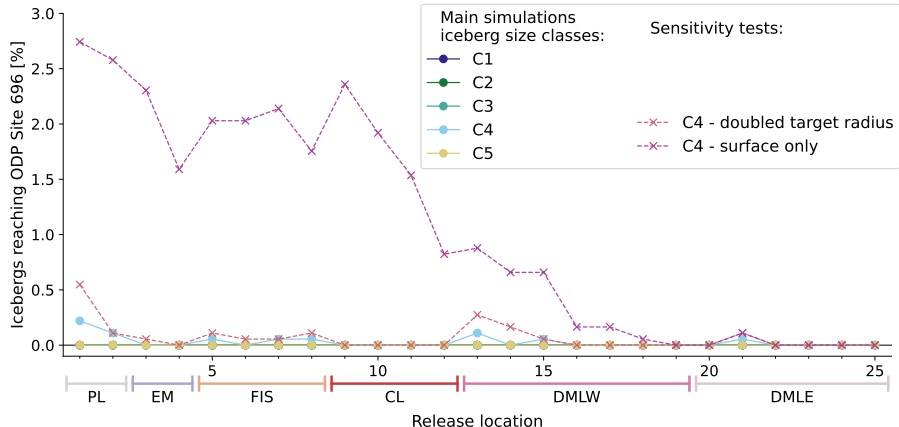

**Figure 4.** Percentage of total releases per forward release location (Fig. 1) that reach ODP Site 696 at some point along their trajectory. Solid lines denote the main forward simulations for each iceberg size class. Dashed lines show the sensitivity simulations of size class C4 when including icebergs passing ODP Site 696 at twice the distance of a grid cell ('doubled target radius') or when using only surface fields ('surface only'). The coastal regions based on Carter et al. (2017) (Fig. 1) are indicated on the $x$-axis where PL = Palmer Land, EM = Ellsworth Mountains, FIS = Filchner Ice Shelf, CL = Coats Land, and DML = Dronning Maud Land (W = west, E = east). The total number and percentage of releases reaching ODP Site 696 per simulation are given in Table A1 (Appendix A).

potential iceberg origins. Then, the results are analysed per defined region to give a clearer view of the most probable, minimum and maximum iceberg size per region.

**Spatial patterns**

The spatial patterns of iceberg mass (Fig. 5a) and thickness (Fig. 5b) clearly show that - irrespective of the size class - iceberg size generally increases with alongshore distance to ODP Site 696. As could be expected, the exact sizes differ strongly

between the size classes, reaching values of over $600\,\mathrm{Mt}$ for C1, $1000\,\mathrm{Mt}$ for C2 and over $15000\,\mathrm{Mt}$ for C3. In terms of thickness, icebergs range between roughly $15$ and $26\,\mathrm{m}$ for class C1, $23$ and $35\,\mathrm{m}$ for class C2, and $205$ to $235\,\mathrm{m}$ for class C3. Interestingly, it seems that the icebergs reaching the region around Gunnerus Ridge are relatively small - that is to say, icebergs here can be smaller or comparable to those reaching the western part of Dronning Maud Land.

A second feature evident from Figures 5a and 5b is that while the spatial extent of icebergs from classes C1 and C2 is quite

similar, icebergs of class C3 can reach a larger part of the Antarctic coast. For class C1 and C2, icebergs reach more or less continuously to $15°$ E and only sporadically further into Dronning Maud Land in the vicinity of Gunnerus Ridge. In both cases, icebergs appear generally unable to reach the southeastern part of the Filchner Ice Shelf and a small patch in front of Coats Land. For C3, coverage is broader and extends until Gunnerus Ridge. Notably, in all three experiments, icebergs can reach the region eastward of Gunnerus Ridge, which did not seem possible in the forward experiment.





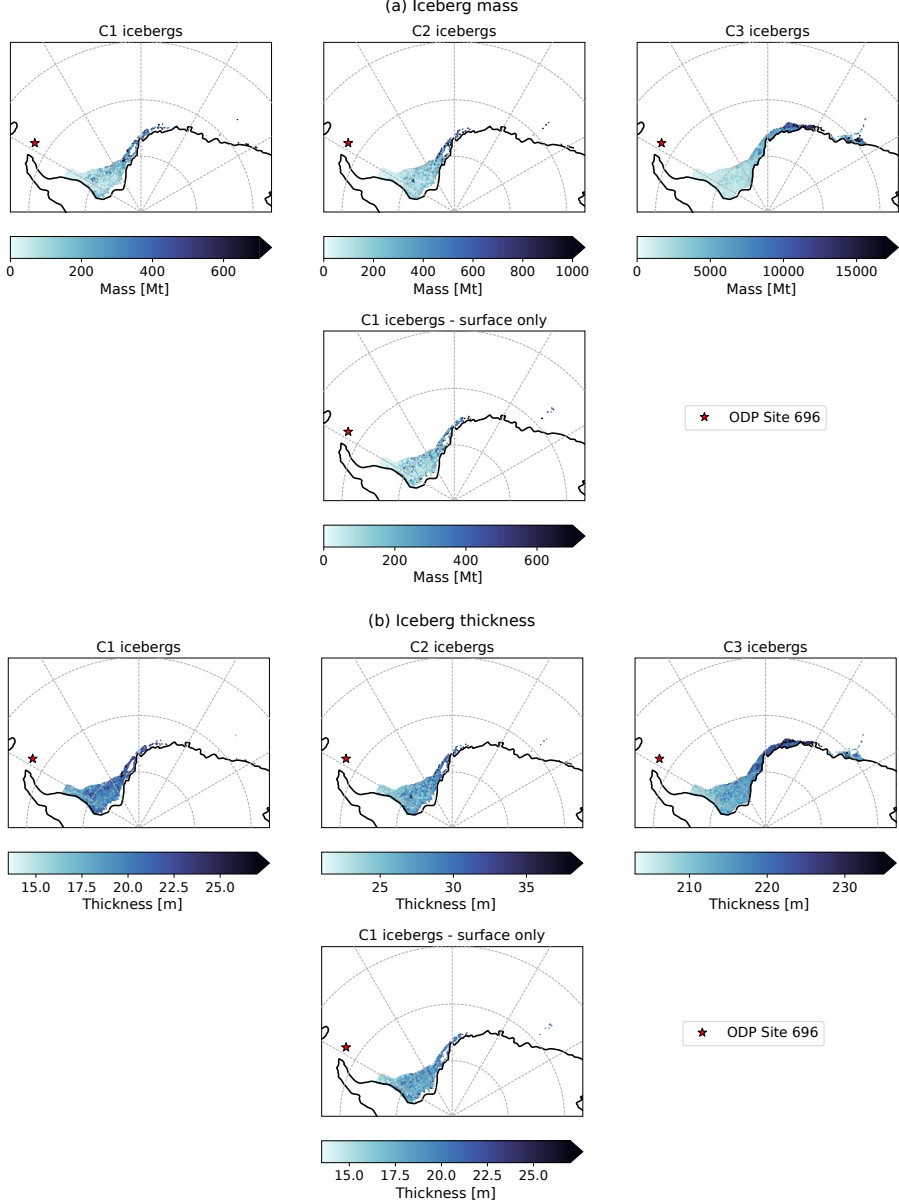

**Figure 5.** Spatial distribution obtained during backward simulations for (a) iceberg mass and (b) iceberg thickness within the defined coastal region of the Southern Weddell Sea based on Carter et al. (2017) (Fig. 1). *Top rows:* results of the main backward simulations ran for iceberg size classes C1, C2 and C3. *Bottom rows:* results of the sensitivity simulation for iceberg size class C1 using only surface fields. The total number of datapoints per simulation are given in Table B1 (Appendix B). Note that the marker of ODP Site 696 is roughly a quarter of the size of the SOM.





## Regional patterns

Turning now to the distribution of iceberg mass (Fig. 6a) and thickness (Fig. 6b) within each coastal region reveals several things. As was apparent in the figures above, both the mass and thickness with the highest probability generally show an alongshore increase through the coastal regions - independent of the initial iceberg size class. The same typically holds for the minimum and maximum values in these regions, but not for the outliers. In addition, the range of probable values of iceberg mass typically increases from Palmer Land to Dronning Maud Land. Again, the order of iceberg size varies between the different size classes. For C1, icebergs able to reach the coast have a mass on the order of $10^{-1}$ to $10^3$ Mt and a thickness of (a few) tens of metres. The range is similar for size class C2, but with the upper limit of iceberg thickness now extended to just above 100 m. For class C3, iceberg mass is on the order of $10^2$ to $10^3$ Mt, with iceberg thicknesses on the order of $10^2$ m.

Going through the size classes one by one, starting at C1, shows a most probable mass of around 50 Mt in Palmer Land, increasing to roughly 400 Mt in the western sector of Dronning Maud Land. In the eastern sector of Dronning Maud Land, iceberg mass is highly variable, with probable values of iceberg mass ranging from around 150 to over 40000 Mt. Interestingly, the range of the minimum iceberg mass observed in the western sector of Dronning Maud Land, including outliers, is smaller than that in any of the other regions and, excluding the western sector of Dronning Maud land, largest in the most proximal region (Palmer Land). Finally, the minimum size observed in Coats Land is larger than that in the more distant western sector of Dronning Maud Land.

For size class C2, overall similar patterns are found with icebergs ranging from just below 10 Mt in Palmer Land to over 50000 Mt in the eastern sector of Dronning Maud Land. Icebergs of size class C3, on the other hand, show some interesting differences. Firstly, the iceberg mass in the eastern sector of Dronning Maud Land is now not only much more constrained and closer to that of the other regions (between roughly 2500 and 25000 Mt), but the most probable mass (around 5000 Mt) is lower than that in the western sector of Dronning Maud Land (10000 Mt). Moreover, the minimum mass (excluding outliers) in the eastern sector of Dronning Maud Land is smaller than that in the western sector (2500 compared to 4000 Mt).

Finally, as was stated before, similar trends are observed in iceberg thickness. However, in addition, Figure 6b shows that the difference in the most probable minimum iceberg thickness between adjacent regions is within roughly 3 m for C1, 4 m for C2 and 10 m for C3. Generally, the difference in thickness between the same region for C1 and C2 varies between 6 to 8 m, and between roughly 185 and 195 m between C2 and C3. In all three cases, the thickness of icebergs in the eastern sector of Dronning Maud Land shows a (slightly) bimodal pattern. For C1 and C2, respective dominant thicknesses lie around 20 and 90 or 30 and 110 m. For class C3, the probabilities are highest around 212 and 222 m. For none of the simulations does the maximum iceberg thickness, including outliers, extend above 260 m.

## 3.4 Model sensitivity

Finally, we are interested in the sensitivity of the model to several variables.



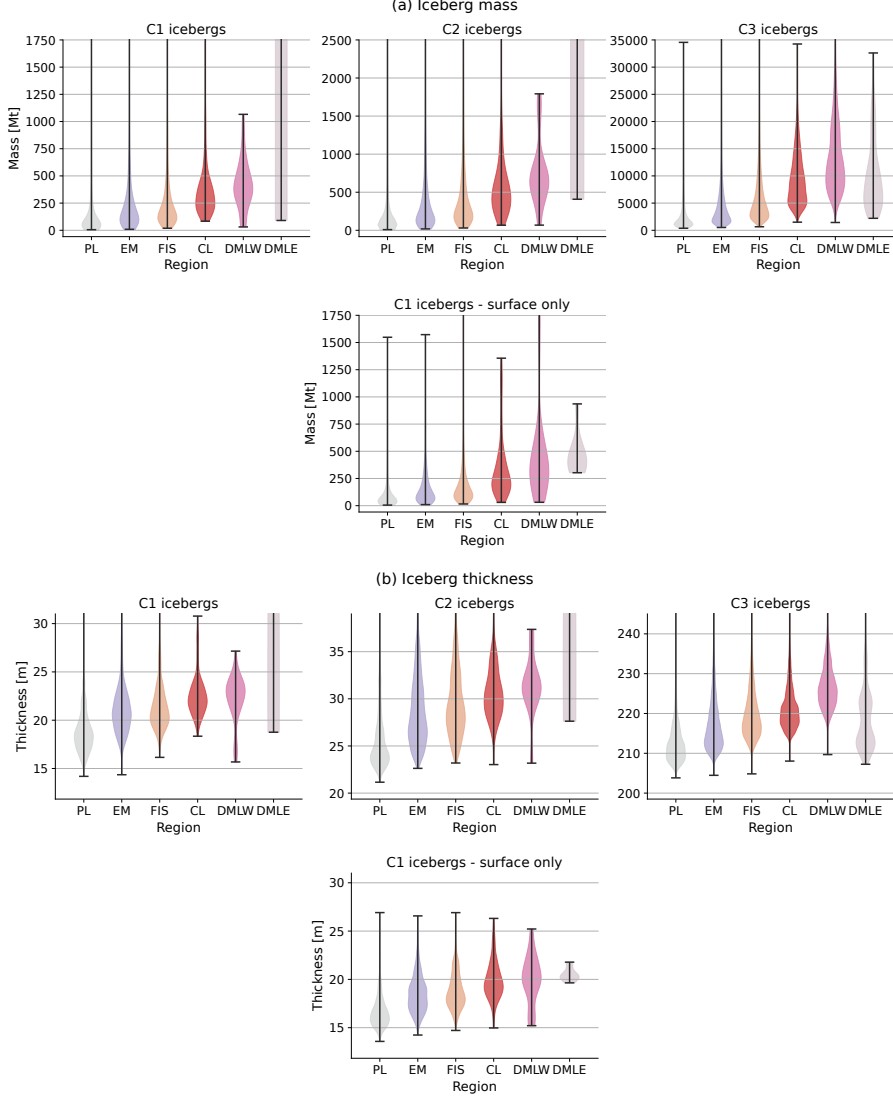

**Figure 6.** Visualisation (violin plot) of the distribution obtained during backward simulations for the (a) minimum iceberg mass and (b) minimum iceberg thickness in each coastal region (Fig. 1) where PL = Palmer Land, EM = Ellsworth Mountains, FIS = Filchner Ice Shelf, CL = Coats Land, and DML = Dronning Maud Land (W = west, E = east). *Top rows:* results of the main backward simulations ran for iceberg size classes C1, C2 and C3. *Bottom rows:* results of the sensitivity simulation ran for size class C1 when using only surface fields. The number of data points per region are given in Table B1 (Appendix B) and the full range of possible values is shown in Figure C1 (Appendix C).

**Doubled target radius**

Including icebergs reaching within the distance of two grid cells from ODP Site 696 increases the number of successful trajectories (Fig. 3). Specifically, whilst for the regular simulation 0.029 % of the trajectories was successful, doubling the





allowed distance increases this to 0.66 % (Table A1 in Appendix A). This increase is twofold. Firstly, several sites previously
not releasing any icebergs to ODP Site 696 are now viable release locations (e.g. locations 3, 6 and 14). Secondly, in addition
to more sites contributing, already successful locations release more icebergs to ODP Site 696, which is especially notable for
release locations 1 and 13 (Fig. 4).

**Surface fields**

Using only surface fields during the simulation for icebergs of size class C4 leads not only to a significant increase in the
number of icebergs reaching ODP Site 696 (Fig. 4) but also to a changed pattern of trajectories (Fig. 3) compared to the
simulation using depth-integrated fields. Icebergs now seem able to travel over the shallower regions along the Antarctic Coast,
such as the Ronne-Filchner Shelf. In addition, some icebergs traverse westwards through the Drake Passage and more icebergs
appear to travel eastward around 57° S. Nevertheless, icebergs released from locations 19, 20, and 22 to 25 in Dronning Maud
Land still appear unable to reach ODP Site 696.

Using only surface fields during backward simulations for size class C1 shows relatively similar results in terms of iceberg
mass (Fig. 6a). In the eastern sector of Dronning Maud Land, however, the iceberg mass is now much more constricted, never
reaching above 1000 Mt. In all regions except the eastern part of Dronning Maud Land, the most probable iceberg thickness is
roughly 2.5 m smaller compared to the depth-integrated simulations (Fig. 6b). As for the iceberg mass, the iceberg thickness
in the eastern part of Dronning Maud Land is more constricted varying between roughly 20 and 22 m.

**4 Discussion**

**4.1 Flow patterns and iceberg trajectories**

An important factor in determining whether Antarctic icebergs could reach ODP Site 696 during the late Eocene is the pattern
of ocean currents. At present, most icebergs released along the Antarctic coast flow with the counterclockwise Antarctic Coastal
Current until meeting the Antarctic Circumpolar Current in the Scotia Sea (Weber et al., 2021), a region colloquially called
'Iceberg Alley'. From both the late Eocene trajectories (Fig. 3) and mean current patterns themselves (Fig. S4.1, Supplementary
Material S4.1.1), we can deduce that a similar current existed along the Antarctic coast during the late Eocene. This is also
supported by other paleoceanographic studies of the Eocene (Huber et al., 2004; Bijl et al., 2011; Sauermilch et al., 2021). In
that regard, the position of ODP Site 696 seems to have been in a location similar to the present-day Iceberg Alley and, as such,
was potentially ideally placed for passing icebergs, if present.

**4.2 IRD provenance**

IRD provenance between Palmer Land and Dronning Maud Land seems possible - albeit sporadically (Fig. 4). Firstly, our
simulations show that trajectories from Palmer Land could reach ODP Site 696, making this a potential source of IRD. Unfor-
tunately, bedrock samples from this region were not taken into account during the analysis by Carter et al. (2017). However,



based on the recently constructed geological map of Antarctica (GeoMAP) (Cox et al., 2023), the mountainous region of
Palmer Land is given an approximately Jurassic to Cretaceous age (200 till 66 Ma), giving it a roughly similar age as North
Graham Land, the region in the northernmost part of the Antarctic Peninsula. Since sediments of the latter region were found
to be relatively dissimilar to those found at ODP Site 696 (Carter et al., 2017), we infer it is unlikely that icebergs from Palmer
Land brought IRD to this site.

To study this further, we can compare these locations to regions of potential glacial formation during the late Eocene, such as
from simulations by Baatsen et al. (2024), who simulated late Eocene Antarctic climate regimes using several climate indices,
and Van Breedam et al. (2022), who used a coupled ice sheet-climate model to study late Eocene glaciations. Although the
extent of glaciation inland differs significantly between these studies, their projected near-coastal glaciated regions match well.
Generally, these studies show that regions at high altitudes and near the coast are likely locations of glacial formation as these
regions are often characterised by lower temperatures, high precipitation rates, or both. Indeed, glaciation is suggested in the
higher altitude regions of Palmer Land (Baatsen et al., 2024; Van Breedam et al., 2022). However, these glaciers were probably
restricted to the higher altitude regions and not marine-terminating, hence inhibiting iceberg production.

A glacial regime is also suggested along the coast between the Ellsworth Mountain and Filchner Ice Shelf regions (Baatsen
et al., 2024; Van Breedam et al., 2022). From the main simulations, however, no successful trajectories arise from the Ellsworth
Mountain region, and only icebergs of size class C4 can reach ODP Site 696 from offshore the Filchner Ice Shelf. Bedrock
samples from this region were also shown to be in good accordance with the provenance of the IRD found at ODP Site 696
(Carter et al., 2017), suggesting this region could be a likely source of the IRD. Whilst rocks from the shelf region of Coats
Land also showed significant similarities to the provenance of the IRD, no successful trajectories in the main simulations exist
for these sites. In addition, neither Baatsen et al. (2024) nor Van Breedam et al. (2022) suggest glacial conditions in this region.
Hence, we can exclude this region as a potential source of IRD.

Finally, icebergs released along Dronning Maud Land can reach ODP Site 696 in our simulations. Although only the western
sector was included in the analysis by Carter et al. (2017), both sectors have a similar geology (Cox et al., 2023) and, as such,
are a potential source of IRD. In addition, the section between roughly 10 and 50° E is shown to have a glacial regime
(Baatsen et al., 2024; Van Breedam et al., 2022). However, depending on the simulation, glacial formation is limited to the
slightly inland high-altitude region only, suggesting it might not have been possible to form marine-terminating glaciers. Still,
we cannot exclude the presence of marine-terminating glaciers and, hence, under the right circumstances, it might have been
possible for icebergs released along the Dronning Maud Land coast to deposit IRD at ODP Site 696 at the SOM.

It is also worth noting that while our study focusses on icebergs reaching ODP Site 696, it seems reasonable to assume
that IRD could have been deposited in a wider area surrounding this site during the late Eocene. For example, as can be seen
from the trajectories in Figure 3, many trajectories seem to visually reach ODP Site 696 but fall outside the used target radius.
As also suggested by the sensitivity simulation with a doubled target radius, increasing this radius even more might lead to
a further increase in the number of successful trajectories. In turn, this might allow iceberg trajectories from the Ellsworth
Mountains region, which was suggested to have a glacial regime (Baatsen et al., 2024; Van Breedam et al., 2022) but did not
provide any successful trajectories with the current target radii used, to be counted as successful.





Finally, in addition to the regions discussed above, one could argue that it might have been possible for icebergs released
west of Drake Passage to reach ODP Site 696 based on the position of South Orkney at the tip of the Antarctic Peninsula during
the late Eocene. Indeed, modelling studies suggest a potential for glacial conditions around $150°$ W in Mary Bryd Land and
at several points along the (western) Ross Sea (Baatsen et al., 2024; Van Breedam et al., 2022). However, Mary Bryd Land
consists predominantly of rocks of Devonian age and younger ($< 420$ Ma) and as such is an unlikely source of the IRD. Whilst
the ages of the bedrock found along the western Ross Sea extend until the Neoproterozoic, no samples older than $760$ Ma have
been found and hence the older part of the age distribution found at ODP Site 696 seems to be absent here too. We thus adhere
to the coastal regions defined by Carter et al. (2017) based on the hinterland geology.

To summarise, the simulated trajectories suggest icebergs released along the coasts from Palmer Land to Dronning Maud
Land can potentially deposit IRD at ODP Site 696, except for the regions offshore the Ellsworth Mountains and the Coats Land
Shelf. However, taking into account potential regions of glacial formation (Baatsen et al., 2024; Van Breedam et al., 2022) and
the local geology (Cox et al., 2023; Carter et al., 2017), the regions offshore the Filchner Ice shelf and Dronning Maud Land
are the most likely source locations of the IRD found at ODP Site 696.

### 4.3 Minimum iceberg size

Having constrained the potential regions from where icebergs could have been released to ODP Site 696 at the SOM, this
section aims to determine the minimum required iceberg size along these stretches of coast and analyse the feasibility of these
sizes.

From the backward simulations starting with icebergs in size class C1, we find that the minimum probable iceberg mass and
thickness offshore the Filchner Ice shelf are around $20$ Mt and $18$ m. The most probable iceberg size in this region is roughly
$125$ Mt and $20$ m. For Dronning Maud Land, the minimum iceberg mass (thickness) following from the simulations is $30$ or
$90$ Mt and $20$ or $19$ m, respectively, with the most probable size roughly $375$ or $45000$ Mt and $23$ or $88$ m.

When starting the backtracking with larger icebergs (class C2), the minimum iceberg mass (thickness) increases to $30$ ($23.5$)
offshore the Filchner Ice Shelf, $150$ ($28$) in the western sector of Dronning Maud Land, and $400$ Mt and $27.5$ m in the eastern
sector of Dronning Maud Land. The most probable minimum size in each region is roughly $250$, $700$, or $45000$ Mt and $27.5$,
$31.5$, or $100$ m. Finally, for icebergs released as size class C3, icebergs originating from offshore the Filchner Ice Shelf should
have a mass (thickness) of at least $900$ Mt ($211$ m), from the western sector of Dronning Maud Land $4000$ Mt ($218$ m), and
from the eastern sector $2250$ Mt ($208$ m). Most probable, however, are icebergs with a mass (thickness) of roughly $2500$ ($215$),
$9000$ ($225$), or $5000$ Mt ($212$ m). In all cases, the sizes obtained at the coast lie roughly above size classes C3 and larger.

In addition to the information on iceberg size gathered from backward simulations, the forward simulations can also provide
an estimate on the order of magnitudes possible. These simulations showed that when icebergs are too small at release from
the coast (size class C1, C2 or C3), they cannot reach ODP Site 696. Only icebergs of size class C4 were able to reach this site
from offshore the Filchner Ice shelf and Dronning Maud Land. Moreover, when icebergs grow too large (C5), they are unable
to reach the site due to interactions with the bathymetry and changes in their trajectories (Fig. D1, Appendix D) resulting in a
lower number of successful trajectories than for class C4 (Fig. 3 and 4). In this case, apparently, the icebergs cannot melt fast




enough to enter water depths around $250\,\mathrm{m}$ - which is the approximate depth of ODP Site 696 during the Eocene. Consequently, the forward simulations suggest icebergs should be around size class C4 upon release at the coast.

Before delving more into the magnitude of the iceberg sizes obtained, we first examine the wide range of probable iceberg sizes obtained in the eastern sector of Dronning Maud Land for classes C1 and C2 (Fig. C1 in Appendix C). For all three size classes from the backward simulations, the relative difference in the minimum mass or thickness in the eastern sector of Dronning Maud Land between the other regions is in a similar range. However, the maximum probable size lies far outside the range of observed sizes, including outliers, for class C1 and C2 but seems still within range for class C3. A possible explanation for this might be that there are only a few data points in the eastern sector of Dronning Maud Land (Table B1 in Appendix B), skewing the results to values that might otherwise have been considered outliers. As is visible in Figure C1 (Appendix C), the distribution in this region also does not contain outliers. Of course, while this could explain the relatively large probable values, it does not explain why such large sizes occur in the first place. This is due to iceberg trajectories that circulate through the Weddell Sea and nearby regions for a long time before eventually reaching the Antarctic coast, consequently allowing them to acquire a sizeable mass.

Returning to the obtained iceberg sizes to analyse whether they are realistic, we start by comparing them to present-day (Antarctic) calving-size distributions, such as that of Stern et al. (2016), which defines ten size classes based on observations. This shows that the minimum iceberg sizes found in this study are on the larger end of regularly found iceberg sizes of the present-day, falling into size classes 7 and larger. In addition, one needs to take into account that the backward simulations give an estimate of iceberg size on the lower side as, for example, grounding has not been taken into account. Hence, it is likely that icebergs would have needed to be larger, falling into even larger size classes from the present-day distribution. Note, however, that icebergs larger than class 10 do exist at present but are considered to calve more infrequently.

To analyse the upper off-scale end of the iceberg classes in more detail, we can compare the obtained iceberg sizes to some of the largest icebergs observed. These include Iceberg B-15 calved from the Ross Ice Shelf, the largest iceberg observed by satellites with an estimated mass of several $10^5$ Mts (Martin et al., 2007) or Iceberg A-68 calved from the Larsen C Ice Shelf with a mass of roughly $10^6$ Mt (Benn and Åström, 2018). It is thus possible at present to form icebergs larger than the minimum sizes suggested by the late Eocene simulations. Nonetheless, such calving events are infrequent and occur at floating ice shelves (Stern et al., 2016). As the late Eocene ocean and air temperatures were relatively warm around Antarctica, it could be argued that the formation of (large) ice shelves would have been unlikely as recent and future warming is shown to cause thinning and retreat of ice shelves (Meredith et al., 2022). However, note that the Eocene model was run without ice in Antarctica. If ice were present in the late Eocene Antarctica, atmospheric and oceanic conditions might have been different locally and, as such, temperatures might currently be overestimated by the model. Hence, the potential for the formation of glaciers, icebergs and floating ice shelves might be underestimated from the current model.

Finally, the collapse of an ice shelf under high temperatures could lead to the release of several (large) icebergs, such as during Heinrich events (Marcott et al., 2011; Hulbe et al., 2004) and, more recently, for the Larsen B Ice Shelf (Cook and Vaughan, 2010). However, as stated before, such large icebergs were likely unable to reach ODP Site 696 as their size would prohibit them from reaching over the shallow regions of the SOM.



In summary, the simulations here suggest icebergs should have been between at least size classes C3 and C4 when released offshore the Filchner Ice shelf and Dronning Maud Land regions in order to be able to reach ODP Site 696. Compared to
present-day iceberg distributions, these icebergs would be on the larger end of the range but not unfeasible as the high Eocene snow accumulation rates could have allowed a fast discharge of ice towards the coast, hence increasing iceberg calving.

## 4.4 Iceberg melt rates

Although ocean circulation plays an important role in determining whether icebergs could reach ODP Site 696 in terms of connectivity, the iceberg melt rates influence these trajectories and determine whether icebergs can survive long enough to
travel the required distance to the site. For a first impression of iceberg melt rates, we can study the half-life time of the icebergs. In general, iceberg lifetime decreases with size. For the present day, icebergs up to $L = 1000$ m (size class C3 in our study) were found to have a half-life time between two and five years in the colder waters in the proximity of the Antarctic coast and one year once entering warmer waters (Orheim et al., 2023), as cited in Wesche and Dierking (2014). Larger icebergs can have lifetimes up to several decades, depending on their size and the waters they drift through (Wesche and Dierking, 2014).

As temperatures in the late Eocene were much higher even at the Antarctic coast - reaching temperatures around 10 °C (Fig. 1c) compared to (sub)zero temperatures at present (Stewart et al., 2019), we expect much shorter lifetimes of the icebergs here. Indeed, when analysing the lifetime of icebergs of size class C3, we find an average half-life time of two months and observe that all icebergs have melted completely within seven months after their release (Fig. E1a, Appendix E). Note, however, that during the simulations data is stored only once every 30 model days, which might cause a deviation of up to 30 days in the
determination of the iceberg (half-)life time. In addition, as stated before, we must not underestimate how much regional cooling could be induced by the presence of ice in the region, which is currently not included in the models. Hence, the iceberg lifetimes found here might be underestimated.

For size class C4, we find that icebergs that travel relatively fast out of the Weddell Sea with the Antarctic Coastal Current disappear in the warmer ocean after roughly one to three years, depending on the latitudes reached. Icebergs that remain in the
coastal region for a long time, likely due to grounding, can survive for three to four years (Fig E1c, Appendix E). In addition, none of the icebergs survive longer than four years and their average half-life time is just under one year, as such indeed being much shorter than at present (Fig E1b, Appendix E).

However, we are also interested in whether and how the contribution of each melt term varies through space and time. As shown in Figure 7, the total iceberg melt during the late Eocene can be up to almost 25 m d$^{-1}$ for icebergs of class C4, which
is much higher than the melt rates observed during the present day. In all terms, the effect of increasing temperatures with decreasing latitude is visible. Even more, the higher values of basal melt can be observed in regions with stronger currents, such as where the icebergs enter the current flowing eastward through Drake Passage. Still, overall, this term is relatively minor, varying between almost 0 m d$^{-1}$ close to the coast up to roughly 0.4 m d$^{-1}$ at lower latitudes. Buoyant convection is slightly larger, ranging between approximately 0.25 and 0.5 m d$^{-1}$. As expected, wave erosion has the strongest influence, ranging
between 10 and 20 m d$^{-1}$.



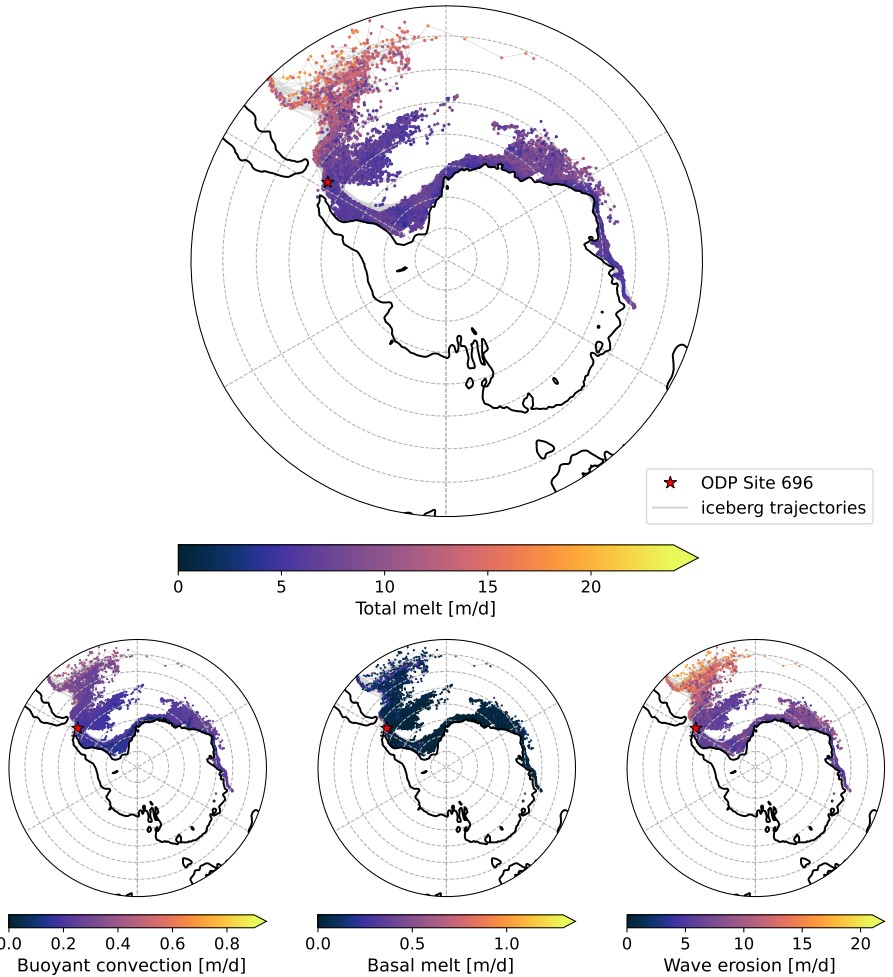

**Figure 7.** Spatial distribution of the total (top) and individual iceberg melt terms (bottom; see Fig. 2) along iceberg trajectories for icebergs of size class C4 during a five-year long forward simulation. Note that the marker of ODP Site 696 is roughly half (equal to) the size of the SOM in the top (bottom) panel(s).

Basal melt still lies within the range of melt rates observed in the present day (Cenedese and Straneo, 2023), although we should note that the magnitude of this term might be underestimated for large icebergs (FitzMaurice and Stern, 2018). The other melt terms - especially wave erosion - are much larger than at present. As buoyant convection is strongly temperature dependent, the higher Eocene temperatures explain the increased melt rate. Similarly, Kubat et al. (2007) suggested that wave erosion can increase with up to $1 \mathrm{~m~d}^{-1}$ per degree temperature. With late Eocene surface temperatures at the Antarctic coast around $10~^\circ\mathrm{C}$ (Fig. S4.1, Supporting Information S4.1), the high magnitude of this term does not seem unreasonable and, taking into account that the modelled temperatures are probably overestimated due to the lack of ice in the model, might even be somewhat lower.





However, as stated before, monthly wind stress data was used to calculate wave erosion. Based on a comparison in the present day (Supporting Information S3.1.4), this could underestimate wave erosion rates by 4 to 46 % (average 31 %). Hence, the late Eocene wave erosion rates might be even larger and, more importantly, the lifetime of icebergs might be overestimated. This could alter the potential provenance regions to ODP Site 696, especially for the more distal regions. Moreover, a change in melt rates might lead to a difference in trajectories. For a modern comparison, this difference appeared to be minor (Fig. S3.5, Supplementary Material S3.1.4). Nevertheless, as this simulation occurred in the much colder present-day setting, which is expected to have lower melt rates, the impact might only become visible after a longer simulation period.

Finally, it should be noted that while the magnitude of melt rates gives a good sense of the impact of environmental conditions on the iceberg, their magnitude is independent of the iceberg's size. Hence, the ratio between the melt terms in units of $\mathrm{Mt\ d}^{-1}$ can show a different contribution of each melt term. Notably, wave erosion only works on two of the iceberg sides (Fig. 2c). For large iceberg masses, the vertical scale of the iceberg (thickness) is usually much smaller compared to the horizontal scale. Therefore, even a small basal melt rate might lead to a large loss of iceberg mass when the horizontal iceberg area is large.

In short, whilst the late Eocene iceberg melt rates are significantly higher than those found nowadays, their magnitude seems fitting for the warm Eocene climate and still allowed larger icebergs to reach ODP Site 696. However, both basal melt and especially wave erosion might be underestimated for (large) icebergs, hence potentially reducing the iceberg lifetimes, which might impact the probable IRD provenances. On the other hand, the lack of ice in the forcing model might lead to an overestimation of temperatures, hence increasing the iceberg's lifetime.

### 4.5 Model sensitivity

Finally, we analyse the sensitivity of the results to several variables.

**Doubled target radius**

Allowing icebergs passing within a larger radius of ODP Site 696 to be included leads to an additional region, the Ellsworth Mountains, able to release icebergs to the site. The more proximal location of this region also slightly reduces the minimum iceberg mass to 100 Mt for size class C1, 150 Mt for C2, and 1750 Mt for C3. The effect on iceberg thickness differs per size class, showing a size increase to 21 m for C1 and a decrease to 26 m and 205 m for C2 and C3, respectively. Overall, the iceberg size appears not to be reduced significantly, and the main effect lies with a change in the possible source locations and, hence, the provenance of IRD. In addition, similar effects might be expected for the other initial size classes (C3, C5).

**Surface-only flow**

From the simulation using only surface fields it appears that all regions might contribute to the IRD at ODP Site 696 to at least some degree. In addition, this seems to lead to slightly smaller most probable minimum iceberg masses and thicknesses compared to the depth-integrated simulation. Apparently, although the higher temperatures at the surface increase the iceberg melt rates - and as such would lead to faster size changes - the increase in iceberg velocity leads to only minor differences



in iceberg size spatially. However, the difference in iceberg trajectories is significant and, as grounding is not included, non-physical.

## 5   Conclusions

This study aimed to simulate late Eocene iceberg trajectories in the Southern Weddell Sea region to test whether these match proxy evidence for iceberg-delivered debris found at ODP Site 696 at the SOM. As stated in the introduction, we determined

that if the IRD is indeed derived from Antarctica, our simulations must demonstrate that 1) icebergs derived from the southern Weddell Sea region can reach ODP Site 696 before having melted away, 2) which regions are most likely to have released icebergs, and 3) what the minimum size of the icebergs leaving Antarctica must be in order to survive the flow path from Antarctica to ODP Site 696.

Our experiments have shown that icebergs released at the Antarctic coast can indeed reach ODP Site 696 before melting

away. Specifically, icebergs derived from offshore the Filchner Ice Shelf and Dronning Maud Land are the most likely source regions for IRD found at ODP Site 696 based on possible trajectories and geological composition. The minimum size of icebergs leaving the coast from these regions must have been on the order of at least $100\,\mathrm{Mt}$ or several 10s of metres in thickness to survive the flow path from Antarctica to ODP Site 696 at the SOM. Although these sizes are at the larger end of the present-day range of common iceberg sizes around Antarctica, the minimum estimates are not unfeasible. Hence, the

present study confirms previous findings suggesting glaciation and iceberg calving were possible in the late Eocene.

A limitation of this study is the use of monthly instead of daily wind fields. A simulation in the present day showed this underestimates wave erosion and, hence, overestimates iceberg lifetime. Further research might explore alternative ways of accounting for this deviation during the simulations. In addition, while expansion of the model by inclusion of other iceberg forcings would improve particularly the trajectories of smaller icebergs, adding a parametrisation of iceberg fracturing will

improve the trajectories of icebergs in the larger size classes. Finally, as grounding was shown to have a significant impact on the iceberg trajectories, simulating different iceberg size classes with various thicknesses could reveal different potential regions of provenance.

*Code and data availability.* The data used in this study was processed using Python 3.8.13. All code used during the simulations and processing of the data is available on GitHub via (Elbertsen et al., 2024a). The model data are available on Zenodo at (Elbertsen et al., 2024b).





## 525  Appendix A: Number of icebergs reaching ODP Site 696

**Table A1.** Total number and percentage of forward iceberg releases reaching ODP Site 696 within one grid cell distance ($\sim 11$ km) for each iceberg size class (C1-C5) and the sensitivity simulations for size class C4 with inclusion of icebergs passing ODP Site 696 at twice the distance of a grid cell ($\sim 22$ km; 'doubled target radius') or when using only surface fields ('surface only').

| Simulation | Icebergs reaching ODP Site 696 | |
|:---:|:---:|:---:|
| | **#** | **%** |
| C1 | 0 | 0 |
| C2 | 0 | 0 |
| C3 | 0 | 0 |
| C4 | 13 | 0.029 |
| C5 | 0 | 0 |
| C4 - doubled target radius | 30 | 0.066 |
| C4 - surface only | 483 | 1.060 |





## Appendix B: Data points per region

**Table B1.** Number of data points in each coastal region (Fig. 1; where PL = Palmer Land, EM = Ellsworth Mountains, FIS = Filchner Ice Shelf, CL = Coats Land, and DML = Dronning Maud Land (W = west, E = east)) and in total for each backward simulation (iceberg size classes C1-C3 and sensitivity simulation of size class C1 using only surface fields).

| Region | Simulation | | | |
|--------|-----|-----|-----|-------------------|
| | C1 | C2 | C3 | C1 - surface only |
| PL | 5043 | 5642 | 24511 | 3896 |
| EM | 3269 | 4332 | 43157 | 1580 |
| FIS | 2494 | 3386 | 32756 | 1453 |
| CL | 350 | 344 | 19348 | 243 |
| DMLW | 65 | 25 | 2638 | 42 |
| DMLE | 8 | 10 | 247 | 9 |
| *Total* | *11229* | *13739* | *122657* | *7223* |



## Appendix C:  Minimum iceberg size per region

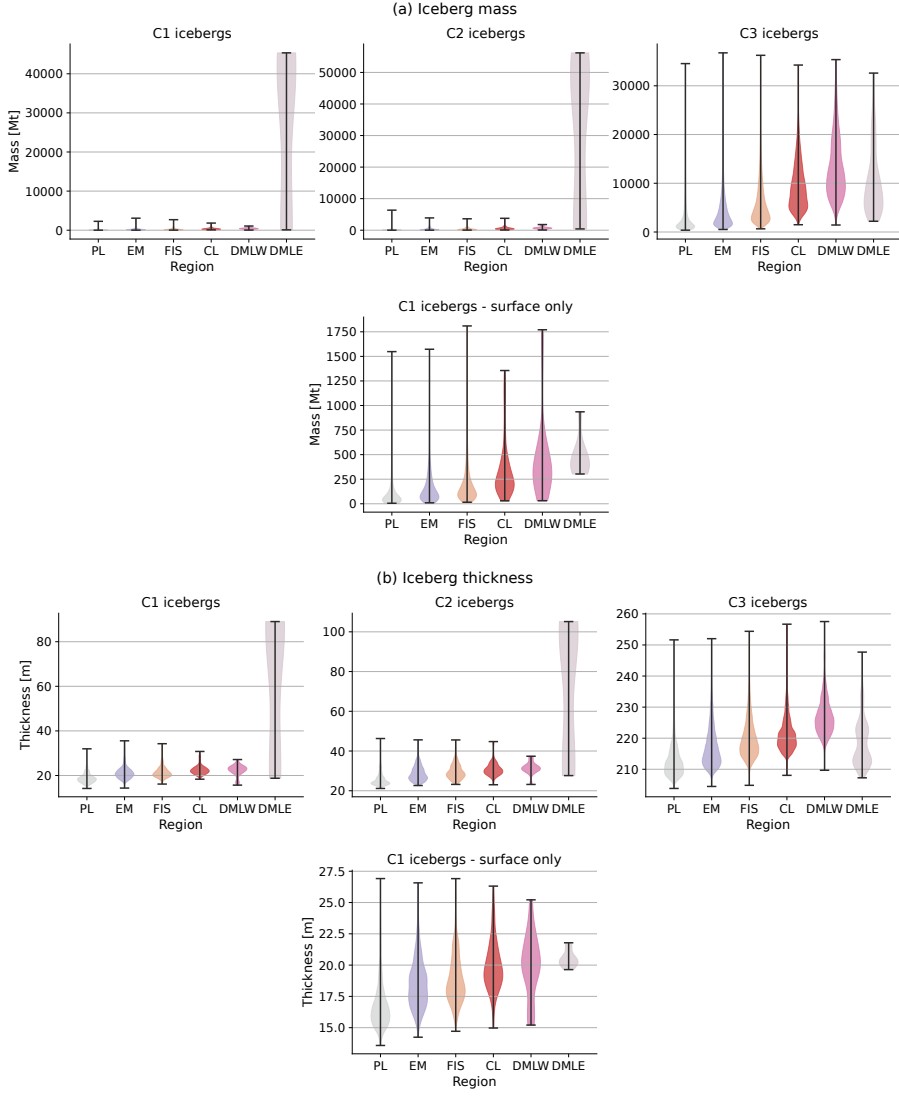

**Figure C1.** Visualisation (violin plot) of the complete distribution obtained during backward simulations for the (a) minimum iceberg mass and (b) minimum iceberg thickness in each coastal region (Fig. 1), where PL = Palmer Land, EM = Ellsworth Mountains, FIS = Filchner Ice Shelf, CL = Coats Land, and DML = Dronning Maud Land (W = west, E = east). *Top rows:* results of the main backward simulations ran for size classes C1, C2 and C3. *Bottom rows:* results of the sensitivity simulation ran for size class C1 when using only surface fields. See Figure 6 for more details in the lower range.



## Appendix D: Iceberg trajectories at the SOM

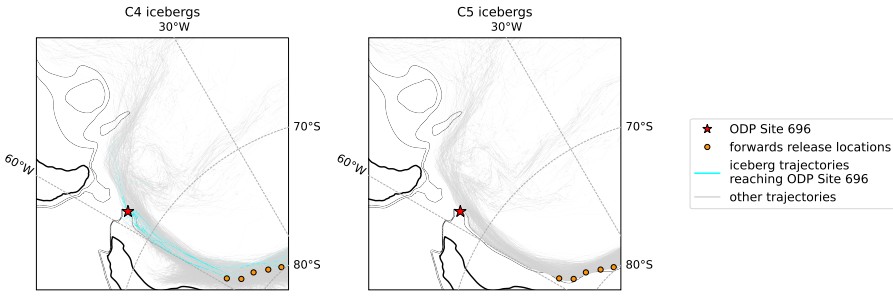

**Figure D1.** Forward trajectories of depth-integrated simulations (iceberg size class C4 and C5) around ODP Site 696 and the SOM that can (blue) or cannot (grey) reach ODP Site 696 at some point along their trajectory. In addition to the coastlines (bold), the 250 and 500 m bathymetry lines are shown. Note that the marker of ODP Site 696 is roughly a sixth of the size of the SOM.





## Appendix E: Iceberg lifetime

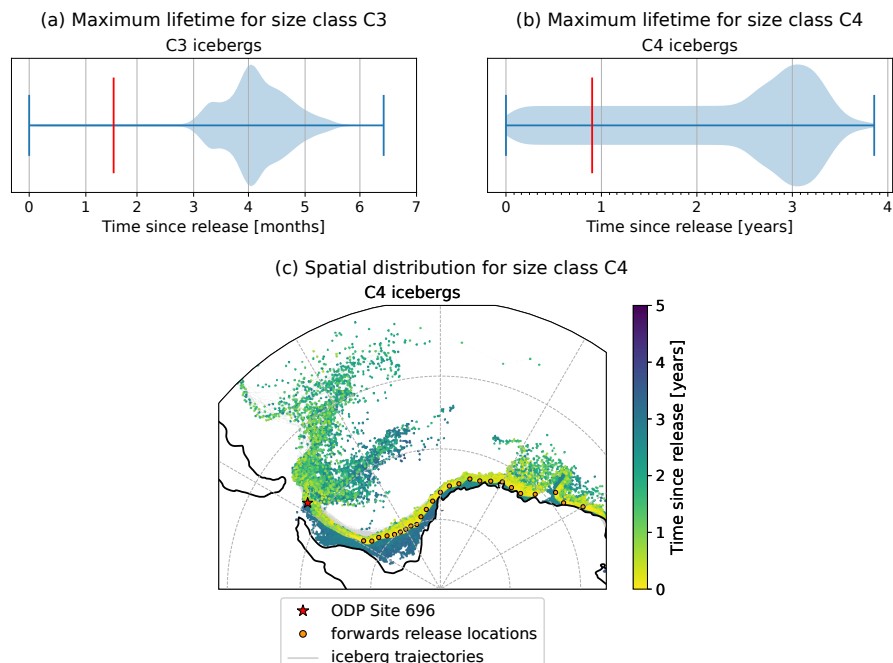

**Figure E1.** Distributions of iceberg lifetime through time (C3, C4) and space (C4) for forward simulations. The red lines in figures (a) and (b) indicate the mean iceberg half-life times. Note that the marker of ODP Site 696 is roughly a quarter of the size of the SOM.



*Author contributions.* PKB and EvS designed the research. MVE and EvS designed the code. MVE ran the simulations and wrote the paper with input of all authors.

*Competing interests.* The authors declare that they have no conflict of interest.

*Acknowledgements.* The authors thank Michael Kliphuis for assisting with and management of the output data. We thank Anna von der Heydt and Peter Nooteboom for providing the model data, and Peter Nooteboom assisting with the model set-up. We also want to thank
Michael Baatsen for providing climate index data.

This research is funded by ERC Starting Grant 802835 (OceaNice) to Peter K. Bijl.




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
