# Peer review of "Possible provenance of IRD by tracing late Eocene Antarctic iceberg melting using a high-resolution ocean model"

_EGUsphere, 2024_

## Author Response (AR1)

**Comments Referee #1**
This paper uses an iceberg trajectory and melting model, embedded within a reasonable late Eocene ocean model to see if it is possible for icebergs from the southern Weddell Sea to have reached an ODP site with Eocene IRD in the South Orkney Islands area. The background and Introduction is well written and argued; it sets the scene well as to what needs to be done to reconcile observations and climate ideas of the time period. The end of the Introduction sets this out well but doesn't strictly say how this paper is tackling the question. Perhaps a synopsis of the paper's structure here would assist the reader? This would be particularly useful as there is very extensive Supplementary Information material.

*Response: We thank the reviewer for the suggestion.*
*Proposed changes: We will add a synopsis at the end of the introduction.*

The methodology section is well and carefully explained. There are a lot of choices made in the model simulations, but the authors argue for their choices well, and are up front about the likely errors in melting rates and trajectories. They do return to the simulation assumptions a number of times, appropriately and sensibly. Overall, there look to be sufficient simulations for the conclusions reached in the paper to be robust and not particularly dependent on the model parameter choices.

Specific questions

l c. 45: What was the palaeo-depth of ODP site 696 during the Eocene? And what is its current depth? It looks shallow in several images for the Eocene – how reliable is the reconstruction?

*Response: Indeed, the paleodepth (~250 m) used in the model is shallower than the present-day depth (~650 m). While the paleodepth cannot be stated with absolute certainty, reconstructions by López-Quirós et al. (2021) do suggest a shallow-water environment persisted during the Late Eocene, followed by a deepening of the SOM from the latest Eocene into the Oligocene. Therefore, the paleodepth used in our simulations is in line with the reconstructed depth range.*

*Proposed changes: We will add this information to the methods (section 2.3).*

Results section

Do they really need the backward trajectory simulations? Surely one can seed the forward simulations with sufficient size ranges to find the minimum sizes needed to be released from the likely origin sites?

*Response: While we could indeed obtain the same information using forward simulations, the fraction of icebergs reaching Antarctica after release from ODP Site 696 in the backward trajectory simulations is much larger than the fraction of icebergs reaching ODP Site 696 from the coast. Hence, using only forward modelling would require running many more simulations and thus cost much more computing time.*

Section 3.1: I think the paper underestimates the likely number of simulations that might provide IRD at the ODP site. There are significant numbers of C4 and C5 iceberg simulations that are shaded grey, reach the South Atlantic, yet don't "score". Given the restrictions on things like advection using monthly mean winds, and the uncertainty from a short ocean simulation window I think the authors are being unreasonably restrictive in forcing only simulations going within 1, or at most 2, gridpoints from the ODP site to contribute to a positive score. To my mind, anything going in the narrow flow near the Orkney sub-continent is a feasible contributor. I encourage the authors to think more flexibly about the errors they allow. The authors do turn to this question in the Discussion section, so it is appreciated by them

*Response: We acknowledge that the range used to select the icebergs reaching the site is conservative, but we believe that increasing the range would not significantly change our main conclusion. As the presented simulations already indicate that icebergs could reach ODP Site 696, a broader range will only increase the likelihood of icebergs reaching the site – as the reviewer pointed out, visually, more trajectories seem to pass the site closely.*

*Proposed changes: We will clarify this more in the methods (section 2.3) and where needed in the discussion.*

Section 3.3: I don't quite understand the backward trajectory studies. Is the iceberg size (ie C1-C3) what size the iceberg is at the ODP site on back release? Does each backward modelled iceberg "grow" through reverse melting processes along its route? This could be explained better in the methodology and probably it is good to remind the reader in section 3.2 as well. Why is the thickness restricted for backward C1 and C2 icebergs? Surely the most likely scenario is for a thick iceberg from a tidewater glacier, or an ice shelf calving and eventually reducing to a few tens of metres when the IRD drops onto the Orkney sub-continent? I can only see forward icebergs being allowed to behave like this, or have I missed something?

*Response: We thank the reviewer for raising these points. The backward simulations are indeed set up in the way described by the reviewer. While the icebergs are influenced by the same processes in both runs (except for the exclusion of grounding during backward simulations), iceberg tipping occurs less often during the backward runs due to the magnitude of the individual melt terms and the surfaces they act on, and this influences the iceberg thickness during backward simulations. However, from running a simulation with size class C4 icebergs without tipping, we can see the overall effect is minor. The trajectories are largely similar to those of the C4 icebergs with tipping, and whilst the maximum lifetime decreases slightly (roughly two months), the effect on the mean iceberg lifetime is small.*

*Proposed changes: We will expand the explanation of the backward simulations in section 2.4 of the methods and remind the reader in 3.3 to emphasise this. We will also place additional context regarding the limited thickness increase during backward simulations compared to forward simulations in the discussion (section 4.3).*

[Figure]

**Figure 1.** Iceberg trajectories and lifetimes for size class C4 with (left) and without (right) allowing iceberg tipping.

510-515: the statement that iceberg thicknesses of a few tens of metres is at the high end of modern estimates for Antarctica must be wrong. Most icebergs today are released from iceshelves hundreds of metres thick. While the authors can argue that Eocene glaciation would probably not have been of this type, even modern tidewater glaciers in the Northern Hemisphere typically produce thicker icebergs than the C1-C2 definitions. The authors need to revisit their wording in places.

*Response: The reviewer is correct. The statement should refer to the order of the iceberg mass.*
*Proposed changes: We will adapt the sentence.*

Table B1 legend needs to be clearer. What is the datapoint variable being shown?

*Response: We agree that the explanation in the legend can be improved to clarify that the data points reflect the number of times any iceberg is located within one of the defined coastal regions during the five-year simulation.*

*Proposed changes: We will clarify the legend of the table.*

A test that is missing, although present implicitly within the various runs, is an attempt to use the geological record for ODP696's IRD to isolate the most likely seed points for the icebergs. This approach might be most revealing and help with the paper's argument. There is currently a lot of detail and simulations that obscures the main message somewhat.

*Response: We thank the reviewer for raising this point. We will clarify our main message more since the question the reviewer poses is what we aim to answer using the simulations presented here.*

*Proposed changes: We will expand the explanation of the locations and regions used in this study in section 2.3 and place additional comments/reminders throughout the text to emphasise the link of the simulations to the IRD evidence of ODP Site 696.*

**Comments Referee #2**

**General Comments**

The authors present intriguing model evidence for late Eocene Antarctic glaciation, via impressive detective work using forward and backward interactive virtual iceberg trajectories. This is achieved through offline use of the Parcels Lagrangian framework extended for icebergs, with data from a simulation of the past ocean circulation that resolves swift narrow boundary currents and some eddying. Considerable thought is given to iceberg size, melting and provenance, with a specific focus on evidence for contemporaneous IRD at ODP Site 696 near South Orkney in the early (narrower) Drake Passage. The Abstract provides a clear and concise summary of findings. Overall presentation is well-structured throughout, figures are informative, and the manuscript is well written. The Supplementary Materials provide extensive technical details, explaining how icebergs are modelled with Parcels, covering specifics of Parcels coding, key equations and parameters, and experimental design. In summary, the manuscript should be suitable for publication, subject to minor revisions in response to the specific comments outlined below.

Specific Comments

1. 9, lines 201-205: Please clarify whether icebergs 'un-melt' (or 're-freeze') along backward trajectories; this is implicit and perhaps obvious given later results and discussion, but it would be helpful to emphasise this here.

   *Response: We thank the reviewer for pointing this out.*
   *Proposed changes: We will expand our explanation of the backward simulations to clarify this in the methods.*

2. 11, Fig. 4: Its appears that C1-C3 (symbols and lines in the legend) are not shown, but I suspect it is simply the case that these symbols and lines are overplotted by identical equivalents for C5; perhaps clarify this in the caption.

   *Response: The reviewer is correct that the lines are overplotted.*
   *Proposed changes: We will add a statement to address this in the figure caption.*

3. 15, Sect. 4.1: Can the authors further remark on how swift/different were the Eocene currents, relative to today, with quantitative evidence from the simulations used here? Any differences in speed and coherence of the Antarctic Coastal Current (ACoC) will impact travel times to Site 696. Around most of Antarctic, the present-day ACoC is largely due to strong (sometimes katabatic) easterly winds which drive a strongly barotropic slope current, modified by buoyancy forcing that introduces some baroclinic shear, all subject to seasonality (the ACoC is stronger in winter). In the region of interest (Weddell Sea), the ACoC is further part of a larger and stronger subpolar gyre, driven by large-scale wind and buoyancy forcing. How might all this be different in the late Eocene simulation, of consequence for iceberg trajectories to Site 696?

*Response: While the current speeds are of the same order (up to roughly 0.4 m/s; see also Fig. 1 below), the ocean patterns are different. This is most obvious in the position and structure of the (proto-) ACC, which shows more latitudinal variation during the Eocene. In addition, this proto-ACC shows a somewhat less continuous meandering current with more eddying behaviour.*

*Regarding the circulation in the Weddell Sea, while the Eocene model shows a narrow region of high-velocities along the Antarctic coast, the flow direction is often different. Although a mostly persistent flow along the Antarctic Peninsula does exist in the Eocene model, it lacks the persistent westward alongshore branch observed in the present-day (see Fig. 2 below). In addition, a larger part of the regions is influenced by onshore flows than at present.*

*The less persistent westward coastal current in the Eocene likely made it more difficult for icebergs released from the further eastward regions to reach ODP Site 696 compared to a present-day current system. However, once an iceberg would get close to the Peninsula, the northward-directed current there would already provide a very suitable path towards the SOM.*

*Proposed changes: We will add additional context in section 4.1 on the (relevant) characteristics of the Eocene model and differences to the present-day. We will add a figure similar to Fig. 1b (and 1c) of the manuscript in the supplements (see figure below) to show the flow (and temperature) field of the present-day model.*

[Figure]

**Figure 1.** Regional time-mean (one year) modern ocean velocity in the surface layer (left) and temperature at the surface (~0-200 m; right).

[Figure]

**Figure 2.** Streamlines of average ocean currents over one year for the Eocene (left) and present-day (right) models used in this study.

4. Further to this, it appears in Figs. S4.1 and S4.2b, and in the animation, that the proto-ACC is narrow and perhaps weaker than today (associated with a narrower Drake Passage and weaker westerlies prior to full Antarctic glaciation?) Is a weakened ACC relative to a 'similar' ACoC (and Weddell gyre) of consequence for icebergs reaching Site 696?

   *Response: It can indeed be assumed that the ACC was not yet fully developed during the late Eocene as the gateways have not deepened and/or broadened sufficiently (see, for example, Stickley et al. (2004) and Scher et al. (2015)). As also pointed out above, there is more latitudinal variation in the position of the proto-ACC. This might have allowed the ACoC to reach further north before deflecting eastward, thus allowing the icebergs closer to ODP Site 696. For example, if one imagines a stronger and more straight ACC in Figure S4.2, this could have led to an increased influence of the proto-ACC relative to the ACoC on the site, thus making it more difficult for icebergs to reach ODP Site 696.*

   *Proposed changes: We will add a short note on the difference in the ACC in section 4.1.*

5. 18, line 420: Can the authors clarify the statement 'However, note that the Eocene model was run without ice in Antarctica'; this statement, along with 'no direct information on sea-ice cover exists in the Eocene model used' (line 145), indicates that the POP model was implemented without a sea ice scheme (this seems odd, given the Los Alamos provenance of POP, shared with the CICE sea ice model); while sea ice may be absent for much of the year, it may be expected to form during mid-winter at high latitudes around Antarctica, when insolation is near-zero – unless substantial heat is transported southward throughout the year. In summary, some more detail on this aspect/assumption of late Eocene high-latitude climate would be appropriate here and elsewhere (see next point).

   *Response: Indeed, the CESM model is coupled to the CICE ice module and runs with a user-defined ice mask. However, for the Eocene, this ice mask does not contain ice. Hence, temperatures around Antarctica are overestimated compared to a simulation where some land ice would be present. Regarding sea-ice, we have briefly discussed the (im)possibility of sea-ice formation around Antarctica in the supplementary material, but we will add a reference back to this in the main text.*

*Proposed changes: We will adapt the text around line 420 for clarification on the absence of land-ice and add refer back to the earlier mention of sea-ice absence to clarify how this could have influenced the simulation.*

6. 19, around line 447: Regarding melting, what about the seasonal cycle? As icebergs are released 'daily at all respective release locations for the five-year model period' (line 206), and that lifetimes vary from months (C3) to years (C4), one might expect that the chance of a given iceberg reaching Site 696 may be sensitive to time of release (through the year). There are some hints at seasonality, e.g., in S4.3 (right middle panel – buoyant convection melt rate) and Fig. S3.5b (wave erosion), so it seems that this may be a factor (e.g., iceberg mass loss by wave erosion is minimal in late summer, so virtual icebergs released in spring will drift further before much melt ensues). How different was late Eocene seasonality (SST, sea ice extent, winds) compared to the present day? Further to this point and related to frequent statements regarding ice (presumably sea ice), it may be relevant that sea ice traps icebergs in winter, which (if winter sea ice were a factor in the late Eocene) may delay multi-year drift time to Site 696.

   *Response: As pointed out by the reviewer seasonal influences seem to be present in some of the melt terms – which we can also expect based on their parameterisations. From other studies (such as Baatsen et al. (2024) and references therein), we know that seasonality on Antarctica was likely amplified during the middle and late Eocene compared to the present day. While the southern hemispheric winters became colder and drier, the summers are characterised by high temperatures and precipitation rates. In that regard, we might expect icebergs released shortly before peak temperatures to have a shorter lifetime than icebergs released shortly after. Regarding sea-ice, we have briefly touched upon the possibility of sea-ice formation around line 145 (section 2.1.3). Based on the SSTs, sea-ice formation seems negligible for any season. However, since the forcing model does not have a dynamic ice component, the SSTs close to the Antarctic coast might be overestimated. Still, this effect will likely be minor compared to the present day.*

   *Proposed changes: We will add a reference to the determination of the absence of sea-ice (section 2.1.3) in the discussion and add a statement to discuss the variation in melt rates through time in reference to Figure S4.3/4.4.*

7. p.19, lines 459-460: Wave erosion is indeed likely to play a dominant role in iceberg mass loss; given the parameterization, Equation (S2.11), and the seasonal cycle (Fig. S3.5b), wind speed is likely to be key to this term. How confident are the authors in the late Eocene winds, irrespective of these being only available from monthly wind stress? How different are late Eocene winds compared to the present day? As explained (lines 165-167), the POP model is 'forced at the surface by a fixed atmosphere of the fully coupled simulations of the Community Earth System Model, or CESM, version 1.0.5 (Baatsen et al., 2020) under a 2 x preindustrial $CO_2$ forcing to simulate the late Eocene'; given this forcing and the late Eocene

geometry, bathymetry and orography, it is likely that the winds are quite different. Can the authors elaborate on this, of relevance to the study?

*Response: A notable difference between the late Eocene and present-day winds in the broader Weddell Sea region is that, especially around Dronning Maud Land, onshore winds persist for a long time of the year (see Fig. 3 below) in contrast to the offshore katabatic winds that are of major importance to a large part of the Antarctic wind patterns in the present day. West of this onshore-wind region, the model shows a region influenced by offshore winds. Specifically, the winds here follow roughly the same path as the icebergs, following the eastern coast of the Peninsula towards ODP Site 696 before turning eastward. In that regard, taking wind drag into account could potentially decrease the time it takes for icebergs to reach the Site.*

*A more detailed but Antarctic-wide description of the Eocene atmosphere and differences to a pre-industrial situation are described by Baatsen et al. (2024) based on simulations using the CESM version 1.0.5 model (4x PIC).*

*Proposed changes: We will add a brief statement to explain the relevance of the different wind patterns during the Eocene.*

[Figure]

**Figure 3.** Mean Eocene wind patterns over a one-year period.

**Comments Referee #3**

In this paper, Elbertsen and co-authors investigate the drift and decay of late Eocene Antarctic icebergs, motivated by the question whether IRD found around the South Orkney Microcontinent may have been transported and deposited by such icebergs. This is a fascinating question and a well-conceived study, with sound rationale to better constrain the potential processes involved. The manuscript is very well written and clearly illustrated. I have a number of mostly minor comments that I hope the authors can consider before publication. As you will see, much of the below is more food for thought than requests for revisions (in particular when I'm referring —with great bias— to my own papers).

General comments:

I appreciated the thorough discussion of limitations and uncertainties. At points, the presentation felt almost too comprehensive, and I was wondering whether the manuscript could be streamlined somewhat. For example, I wonder whether back-of-envelop estimates on drift and decay could be used earlier on to eliminate small icebergs, with draft O(10 m), as viable candidates, and then focus the discussion more on iceberg categories C3-C5? Alternatively, I would suggest that a brief discussion of how icebergs C1-C2 are unlikely to be significant contributors, based on your simulations, can justify leaving them out of the main results?

*Response: We thank the reviewer for the suggestion. We agree that the C1 and C2 class icebergs are the least significant contributors based on our simulations.*

*Proposed changes: We will aim to deemphasise their results throughout the text to some extent but will keep the article structure intact otherwise as it also supports the results from the C3 to C5 iceberg classes.*

- I agree with Reviewer 1 in that I am not 100% clear I follow the argument that the backward simulations are needed (and how these simulations are exactly performed). The fact that these simulations suggest that rather thin icebergs are likely main contributors gives me pause. Two other factors play into this: (1) the impact of winds on small icebergs (which are seeded at ODP 696 for the backward runs) and (2) the major impact of grounding. I'll discuss both a bit more.

*Response: Using backward simulations, we can limit the computing time used as the fraction of icebergs reaching Antarctica after release from ODP Site 696 is much larger than the fraction of icebergs reaching ODP Site 696 from the coast. Regarding the small thickness, here the process of iceberg tipping plays a role as this occurs less often during the backward runs due to the magnitude of the individual melt terms and the surfaces they act on, which favours horizontal iceberg growth. However, from running a simulation with size class C4 icebergs without tipping, we can see the overall effect is minor. The trajectories are largely similar to those of the C4 icebergs with tipping, and whilst the maximum lifetime decreases slightly (roughly two months), the effect on the mean iceberg lifetime is small.*

*Proposed changes: We will add a clearer description of the backward simulations in section 2.4 and extend the explanation of the limited iceberg thickness in the discussion.*

Impact on wind on small icebergs: As the authors acknowledge, this is a major limitation of this modeling framework. We showed in Wagner et al (JPO, 2017) that icebergs smaller than length ~1km are significantly impacted by wind forcing, and it's a long-standing rule-of-thumb that small icebergs roughly drift at 2% of the wind speed, relative to the water. In Fig 7 of that paper you can see how (in a modern climate) smaller icebergs from the Weddell Sea are advected much further east than large icebergs. The assumption that icebergs drift approximately with the ocean currents is well approximated for icebergs > 1km length (Fig 4 of that paper). In modern day climates, >90% of iceberg volume is contained in such large icebergs, and if the late Eocene icebergs broke off from ice shelves, I would expect those also to be predominantly large. And even if the Antarctic coast was dominated by tidewater glaciers with large ice thicknesses, such as the larger modern-day Greenland glaciers which can reach front thicknesses up to 1000 m (see, e.g., Bassis and Walker, "Upper and lower limits on the stability of calving glaciers from the yield strength envelope of ice", Proc R Soc A, 2012), then the large limit may be justified. (I mostly mention this because I'd urge the authors to de-emphasize the C1/C2 simulations).

*Response: Indeed, the model frame used might be less applicable to the smallest icebergs. However, as pointed out, these small classes are also least relevant as their lifetime is too short. In that regard, the larger icebergs are better represented in the model frame and also seem to be the most realistic contributors to the IRD.*

*Proposed changes: We will add a note on the limitation of using only the ocean velocity field as forcing for the smallest size classes in section 2.1.3 and refer back to this in the discussion of the C1 and C2 simulations.*

Grounding: looking at the discrepancy in Fig 4 between C4 icebergs with surface-only versus depth-averaged forcing suggests a major impact of grounding in these simulations. I believe that this is less pronounced in present-day Antarctica. Large icebergs do run aground for long periods (e.g., A23a), but they don't deteriorate much on the sides because they are largely protected by sea ice in the Weddell Sea. So they gradually thin and then continue the journey once their keel is shallower than the ocean depth - the main impact on grounding is then a delay in their trajectory. I wonder whether the late Eocene situation may have been somewhat similar, in particular given the question of whether there may have been some sea ice in the Weddell Sea (see comment on sea ice by Reviewer 2)? We once made the argument that sea ice can help expand the spread of IRD (Wagner et al, 2018, "Wave inhibition by sea ice enables trans-Atlantic ice rafting of debris during Heinrich events") although I'm not sure Heinrich events are a good proxy for the late Eocene. Finally, even if there was no sea ice, I wonder whether grounding times are biased high in the present simulations? Basal melt rates scale with the velocity difference between the ice and the ocean, so for grounded icebergs this should be elevated. I assume the simulations take this into account, but I was surprised by how big the impact of grounding seems to be (and the basal melt rates in Fig 7 are pretty small).

*Response: As the high-resolution model does not contain a sea-ice component, we looked at the possibility of sea-ice formation based on sea-surface temperatures (around l.145 in the original manuscript). As these were generally too high, we assume the effect of sea ice on icebergs was negligible. Regarding the effect of iceberg grounding, one of the factors making it more likely for icebergs without grounding to reach ODP Site 696 is that while indeed basal melt rates are elevated for grounded icebergs, they are still low compared to the other melt terms. Hence, even a short grounding interval can lead to a quite strong reduction in the horizontal dimensions of the iceberg, making it melt away before reaching the site and/or altering its trajectory.*

*Proposed changes: We will remind the reader on the sea-ice absence and elaborate on iceberg grounding and melt rates in section 4.4.*

- I think it's important to highlight that the melt model used here, which originated by Bigg et al (1997), was designed to match the deterioration of small (Arctic) icebergs. As you discuss in 2.1.3., the decay of larger icebergs really should include a representation of breakup (explicit or implicit), otherwise you get unrealistically long survival (see Rackow et al 2017). Relatedly, the iceberg size distributions of the Antarctic studies that built on the work of Bing et al (1997), namely Gladstone et al (2001), Martin & Adcroft (2010), Stern et al (2016) all only consider icebergs that are too small, compared to real-world Southern Ocean icebergs. I would revise the statement on l.407 to reflect that the Stern et al size classes are only capturing the small icebergs.

*Response: We agree that it is useful to the reader to clarify this statement to ensure the intentions behind and the limitations of the size classes and melt model are clear.*

*Proposed changes: We will revise this paragraph in the discussion.*

- Validation: As an upshot of the main model limitations (in my view the lack of wind forcing on small icebergs and the lack of breakup representation on large icebergs), I do wonder whether it may be worth trying to do some form of validation? Do you have any simulations with the same ocean model but for present day conditions, which you could seed with icebergs to see the match with present-day iceberg distributions? Alternatively, one could perform some rough validation runs with an idealized model (like the one we developed in Wagner et al, JPO, 2017), by using your size classes and turning wind forcing on and off to check the impact this has on the general distribution. I appreciate this is beyond the scope of the present study and more intended as ideas for next steps.

*Response: We thank the reviewer for these suggestions. We have briefly considered the possibility of validating the model with present-day simulations but decided it was beyond the scope of this study.*

Minor:

- l. 69 I would argue that you do take Coriolis into account (the ocean currents are determined by Coriolis). In Wagner et al (JPO, 2017) we show that the iceberg drift converges to that of the ocean currents in the large-iceberg limit, and we retain the Coriolis force throughout this analysis.

  *Response: It is good that this is pointed out. Indeed, the Coriolis force is indirectly forcing the icebergs through the ocean currents in our model. Our aim here was to point out that we do not take the direct influence of the Coriolis force on icebergs into account. As the reviewer correctly states, the effect of this is probably minor as most of our simulations use sufficiently large icebergs where their movement can be described based on the ocean currents.*

  *Proposed changes: We will make the distinction between the direct and indirect Coriolis force clearer in this sentence.*

- I agree with the other Reviewer that your target area seems too restrictive (given the large uncertainties throughout)

  *Response: We acknowledge that the range used to select the icebergs reaching the site is conservative, but we believe that increasing the range would not significantly change our main conclusion. As the current simulations already indicate that icebergs could reach ODP Site 696, a broader range will only increase the likelihood of icebergs reaching the site – as pointed out, visually, more trajectories seem to pass the site closely.*

  *Proposed changes: We will clarify this more in the methods (section 2.3) and where needed in the discussion.*

- l.115 This makes it sound like basal melt is a main driver of deterioration, but it is typically (in my experience) an order of magnitude smaller than wave erosion (which is also in line with your Fig 7). I am aware that Bigg (2015) describes it as a main driver, but I do think 0.1 m/day is a more typical scale, rather than 1 m/day (except maybe for high differential velocities when icebergs are grounded, see above).

  *Response: We understand how this sentence can give the wrong impression about the dominant melt process.*

  *Proposed changes: We will rephrase this to better explain the potential and common ranges of the melt rates.*

- L.125 It's somewhat unclear to me whether depth-integrated ocean properties are an improvement over surface-only properties for the wave erosion calculation (since this is determined by the near-surface water temperatures). However, the wave erosion term is so ill-constrained in general that I would assume this to be a small dial to turn.

*Response: Following Merino et al. (2016), the parametrisation for wave erosion indeed still depends on the properties at the atmosphere-sea interface and not on depth-integrated properties.*

*Proposed changes: We will add a short statement in section 2.1.2 to clarify this.*

- I will just note that we looked at the impact of winds on drift and deterioration in some detail in Wagner & Eisenman ("How climate model biases skew the distribution of iceberg meltwater", GRL, 2017). I bring this up because we found a compensating effect when wind speeds increase: the decay rate goes up, but so do iceberg drift speeds, and as a result the distribution of melt water stays roughly the same (in that sense, biases in wind speeds may not be that impactful on meltwater distribution).

*Response: We thank the reviewer for bringing this up. The compensating effect referred to is indeed relevant to our study as it might suggest that the impact of the potential biases in the wind speeds on the iceberg lifetimes might be limited. We will keep this in mind during the revision.*

- l.294, increase for two reasons, not a "twofold increase"

*Response: We thank the reviewer for pointing this out.*
*Proposed changes: We will adapt this sentence.*